# Capsular polysaccharide restrains type VI secretion in *Acinetobacter baumannii*

**Nicolas Flaugnatti[1], Loriane Bader[1], Mary Croisier-Coeytaux[2], Melanie Blokesch[1]***

[1]Laboratory of Molecular Microbiology, Global Health Institute, School of Life Sciences, Ecole Polytechnique Fédérale de Lausanne, Lausanne, Switzerland; [2]Bioelectron Microscopy Core Facility, School of Life Sciences, Station 19, EPFL-SV-PTBIOEM, Ecole Polytechnique Fédérale de Lausanne, Lausanne, Switzerland

**Abstract** The type VI secretion system (T6SS) is a sophisticated, contact-dependent nanomachine involved in interbacterial competition. To function effectively, the T6SS must penetrate the membranes of both attacker and target bacteria. Structures associated with the cell envelope, like polysaccharides chains, can therefore introduce spatial separation and steric hindrance, potentially affecting the efficacy of the T6SS. In this study, we examined how the capsular polysaccharide (CPS) of *Acinetobacter baumannii* affects T6SS's antibacterial function. Our findings show that the CPS confers resistance against T6SS-mediated assaults from rival bacteria. Notably, under typical growth conditions, the presence of the surface-bound capsule also reduces the efficacy of the bacterium's own T6SS. This T6SS impairment is further enhanced when CPS is overproduced due to genetic modifications or antibiotic treatment. Furthermore, we demonstrate that the bacterium adjusts the level of the T6SS inner tube protein Hcp according to its secretion capacity, by initiating a degradation process involving the ClpXP protease. Collectively, our findings contribute to a better understanding of the dynamic relationship between T6SS and CPS and how they respond swiftly to environmental challenges.

*For correspondence:
melanie.blokesch@epfl.ch

**Competing interest:** The authors declare that no competing interests exist.

## Editor's evaluation

This important study reveals that capsular polysaccharide (CPS) in Acinetobacter baumannii not only protects against type VI secretion system (T6SS) attacks but also impairs the bacterium's own T6SS efficacy. The evidence supporting these findings is compelling. This work will be of interest to researchers focusing on bacterial defense mechanisms and interbacterial competition.

## Introduction

*Acinetobacter baumannii* is an opportunistic pathogen known for causing hospital-acquired infections. The World Health Organization (WHO) has identified it as a critically high-priority pathogen in dire need of new therapeutic strategies (*Tacconelli et al., 2018*). Consistent with its classification, *A. baumannii* is a member of the 'ESKAPE bugs', a term referring to six pathogenic species (*Enterococcus faecium*, *Staphylococcus aureus*, *Klebsiella pneumoniae*, *Acinetobacter baumannii*, *Pseudomonas aeruginosa*, *Enterobacter* spp.) notorious for causing hospital-acquired infections and their ability to escape antibiotic treatments (*Rice, 2008*).

A. baumannii can gain new functions, including antibiotic resistance, through horizontal gene transfer, notably via plasmid conjugation (*Di Venanzio et al., 2019*; *Hamidian et al., 2014*) and natural competence for transformation (*Godeux et al., 2018*; *Godeux et al., 2022*; *Harding et al., 2013*; *Ramirez et al., 2010*; *Vesel and Blokesch, 2021*; *Wilharm et al., 2013*). Beyond its resistance to antibiotics, the bacterium can withstand desiccation, disinfectants, and survive on surfaces

for extended periods, posing a significant challenge in hospital environments (*Harding et al., 2018*). For *Acinetobacter baylyi*, a non-pathogenic species belonging to the same genus, resilience against external stresses was shown to be at least partly attributed to extracellular polysaccharides (*Ophir and Gutnick, 1994*). These polysaccharides, whether secreted into the environment as biofilm matrix components (exopolysaccharides; EPS) or part of/attached to the bacterial membrane (like lipopoly-saccharide, lipooligosaccharide, and capsule [capsular polysaccharide; CPS]), serve various protective roles against physical, chemical, and biological stresses (*Flemming et al., 2023*; *Paczosa and Mecsas, 2016*; *Simpson and Trent, 2019*; *Whitfield et al., 2020*). In *A. baumannii*, the capsule, encoded within the genomic region between the *fkpA* and *lldP* genes known as the K locus (*Wyres et al., 2020*), is a key feature for many strains. The K locus is usually arranged in three parts: (1) the genes encoding the CPS export machinery (e.g., the *wza*, *wzb*, *wzc* operon); (2) a central region for capsule construction and processing (including the genes *wzy* and *wzx*, which encode the repeat unit polymerase and translo-case, respectively); and (3) a module for synthesizing simple sugar substrates (*Wyres et al., 2020*). The CPS assembles into complex, multibranched glycans that are tightly anchored to the outer membrane by the Wzi protein, effectively encasing the cell in a protective polysaccharide shield (*Tickner et al., 2021*). This capsule plays a crucial role in the virulence of *A. baumannii*, as demonstrated in in vivo studies using animal models where it provided resistance against complement-mediated killing (*Lees-Miller et al., 2013*; *Russo et al., 2010*). Furthermore, monoclonal antibodies targeting CPS have been shown to protect mice from infection by hypervirulent strains (*Nielsen et al., 2017*). Despite its significant role in *A. baumannii* pathogenicity, the regulatory mechanisms governing capsule produc-tion remain largely unexplored. However, recent findings indicate that exposure to specific antibiotics at sub-MIC concentrations, such as of chloramphenicol, can trigger the upregulation of K locus genes (*Geisinger and Isberg, 2015*). This response is mediated by the BfmRS two-component regulatory system, leading to enhanced virulence of *A. baumannii* (*Geisinger and Isberg, 2015*).

Extracellular polysaccharides such as EPS and CPS are known to protect against attacks by bacteria with a type VI secretion system (T6SS), a key player in bacterial warfare (*Smith et al., 2023*). Found in 25% of Gram-negative bacteria and in more than 50% of β- and γ proteobacteria (*Abby et al., 2016*; *Bingle et al., 2008*), the T6SS is a contact-dependent contractile machinery that resembles inverted contractile bacteriophage tails (*Basler et al., 2012*; *Leiman et al., 2009*). The T6SS features a membrane complex that extends across both the inner and outer bacterial membranes, with a baseplate-like structure connected within the cytoplasm (*Cherrak et al., 2018*; *Durand et al., 2015*). The baseplate of the T6SS is linked to an internal tube composed of Hcp hexamers, encased by a contractile sheath formed by TssB and TssC proteins (*Basler et al., 2012*). Upon contraction, the T6SS propels its inner tube, along with VgrG-PAAR spike proteins and toxins, into neighboring cells, leading to either growth arrest or cell death (*Cherrak et al., 2019*; *Russell et al., 2014*). To protect themselves from the effects of their own T6SS-launched toxins, bacteria that possess the T6SS also produce immunity proteins. These proteins specifically neutralize the bacterium's own toxic effector proteins, preventing self-intoxication or intoxication of kin (*Hood et al., 2010*; *MacIntyre et al., 2010*; *Russell et al., 2011*). As mentioned above, recent research has identified mechanisms of resistance to T6SS toxicity that do not involve immunity proteins, which include defenses provided by the production of EPS (*Granato et al., 2023*; *Hersch et al., 2020*; *Toska et al., 2018*) and capsules (*Flaugnatti et al., 2021*).

The T6SS is widely found across *Acinetobacter* species, including *A. baumannii* (*Dong et al., 2022*; *Weber et al., 2013*). The genes encoding the core components of the T6SS reside in a single locus. However, the *vgrG* genes, which are crucial for the system's function, are scattered throughout the chromosome alongside effector/immunity modules (*Eijkelkamp et al., 2014*; *Lewis et al., 2019*).

The regulation of the T6SS in *A. baumannii* varies, with some isolates expressing the system under standard laboratory conditions, while others regulate expression via proteins such as H-NS (*Eijkelkamp et al., 2013*; *Repizo et al., 2015*; *Weber et al., 2013*). Additionally, TetR-like repressors encoded on large conjugative plasmids, which also bear antibiotic resistance genes, can suppress T6SS to aid conjugation and plasmid dissemination among cells (*Di Venanzio et al., 2019*; *Weber et al., 2015*). This diversity in regulatory mechanisms indicates a complex interplay between antibiotic resistance, T6SS activity, and bacterial competitiveness. Indeed, the T6SS in *A. baumannii* significantly influences interbacterial dynamics, effectively targeting not only Gram-negative and -positive bacteria (*Le et al., 2021*; *Weber et al., 2013*) but also exhibiting antifungal capabilities (*Luo et al., 2023*).

However, although the T6SS confers competitive advantages to *A. baumannii* by targeting a wide range of microorganisms in vitro, studies in diverse animal models have shown that T6SS mutants do not incur a fitness cost (*Weber et al., 2013*), a finding that might be strain dependent, at least in the *Galleria mellonella* wax moth model of disease (*Repizo et al., 2015*). This suggests that the T6SS's role in the bacterium's virulence might not be direct (*Subashchandrabose et al., 2016*; *Wang et al., 2014*), highlighting a complex interaction with host organisms and/or its environment that warrants further investigation.

In this study, we investigate the impact of capsule production on T6SS antibacterial activity in *A. baumannii*. Our findings reveal that the CPS acts as a shield against T6SS attacks from rival bacteria. Despite this, many *A. baumannii* strains also have an operational T6SS, underscoring the capsules primary role as a one-way barrier. However, we show that under typical laboratory growth conditions, the presence of the surface-bound capsule nonetheless reduces the efficacy of the bacterium's own T6SS. This T6SS impairment is further enhanced when CPS is overproduced due to genetic modifications or antibiotic treatment. Finally, we go on to demonstrate that when T6SS secretion is hindered, the accumulation of Hcp protein components is curtailed by a degradation process facilitated by the ClpXP protease system.

## Results and discussion

### The capsule of *A. baumannii* contributes to protection against T6SS attacks

In our study, we explored how capsule production influences the antibacterial activity of the T6SS in *A. baumannii*, specifically focusing on the clinical isolate A118 (*Ramirez et al., 2010*; *Traglia et al., 2014*). The A118 strain was found to possess the K locus, located between the *fkpA* and *lldP* genes (*Vesel and Blokesch, 2021*), suggesting its encapsulated nature. To confirm capsule production, we created a mutant lacking the *itrA* gene (*Bai et al., 2021*), essential for the initial steps of glycan chain formation (*Kenyon and Hall, 2013*). Analysis of the CPS material revealed the presence of high molecular weight polysaccharide in the wild-type (WT) strain but not the ΔitrA mutant, as indicated by Alcian blue staining (*Figure 1A*). Next, we challenged both strains with rabbit serum to assess complement-mediated killing. As shown in *Figure 1B*, deleting the *itrA* gene resulted in a three-log decrease in survival compared to both the WT strain and the mock control conditions. This complement-dependent killing of the non-capsulated strain was not observed when the serum was heat-inactivated before being added to the bacteria (*Figure 1B*). Collectively, and in accordance with current literature (*Kenyon and Hall, 2013*; *Lees-Miller et al., 2013*), our findings establish the critical role of ItrA in CPS synthesis, and confirm that *A. baumannii* A118 possesses a capsule.

In our previous work, we observed that *A. baumannii* exhibited minimal susceptibility to *Enterobacter cloacae*'s T6SS (*Flaugnatti et al., 2021*), hinting at the potential protective role of its CPS against T6SS-mediated attacks. To investigate this protection further, we conducted a killing assay using capsulated (WT) and non-capsulated (ΔitrA) *A. baumannii* strains as prey in a T6SS-inactivated (T6SS−) strain background. Interestingly, the attacker *E. cloacae* was ineffective in killing CPS-producing *A. baumannii* prey (WT carrying a cargo-less transposon; WT-Tn) in a T6SS-dependent manner (*Figure 1C*). Conversely, the lack of CPS in the *A. baumannii* prey (ΔitrA-Tn) resulted in increased susceptibility to T6SS-mediated attacks by *E. cloacae* (*Figure 1C*). This vulnerability could be reversed by introducing a new copy of the *itrA* gene into the strain's genome (ΔitrA-Tn-itrA), as depicted in *Figure 1C*. When we conducted the experiment again under conditions that allow *A. baumannii*'s T6SS to function (T6SS+), we observed an unexpected outcome: the non-capsulated strain (ΔitrA) exhibited full resistance to *E. cloacae*'s T6SS attacks, mirroring the resistance shown by the capsulated WT strain (*Figure 1D*). Indeed, several studies have reported that certain isolates of *A. baumannii* are equipped with a constitutively produced antibacterial T6SS (*Repizo et al., 2015*; *Weber et al., 2013*). To confirm the functionality of the T6SS in strain A118, we disrupted either the *hcp* or the *tssB* gene within the strain's main T6SS gene cluster (*Figure 1—figure supplement 1A*), which encode essential components of the system, and assessed the impact on T6SS-mediated killing activity against *E. coli* prey (*Figure 1—figure supplement 1B*). As expected, removing *hcp* and *tssB* effectively eliminated the T6SS's antibacterial capabilities. Additionally, experiments with selectable *E. cloacae* (*Figure 1—figure supplement 1C, D*) confirmed that *E. cloacae* is killed by *A. baumannii*

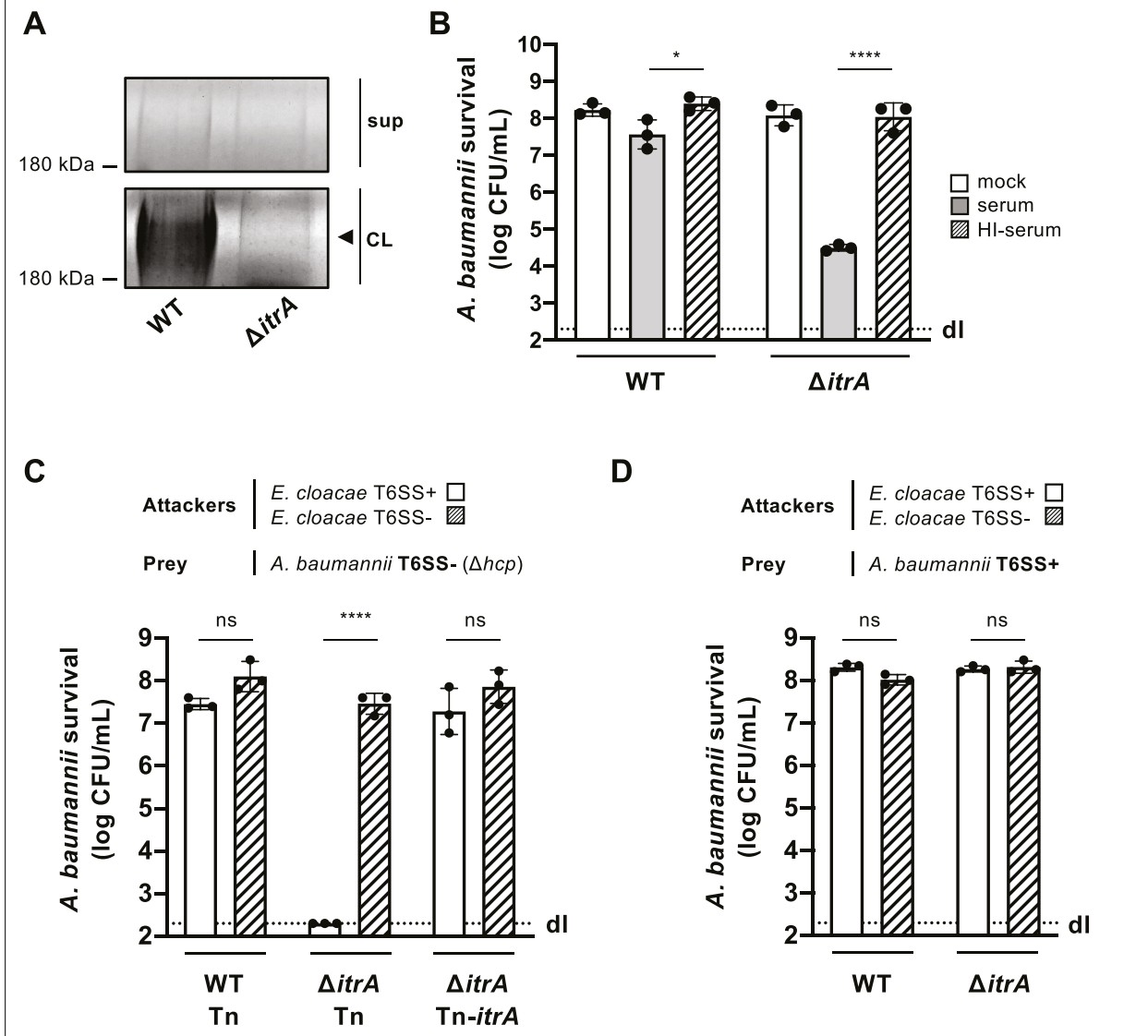

**Figure 1.** Capsular polysaccharide protects *A. baumannii* against external T6SS assaults. (**A**) Analysis of polysaccharides in the cell lysate (CL) and supernatant (sup) of wild-type (WT) or capsule-deficient (Δ*itrA*) strains of *A. baumannii,* separated by SDS–PAGE and stained with Alcian blue. The arrowhead indicates the polysaccharide band. (**B**) Protection against complement-mediated killing. Exponential growth cultures of WT and Δ*itrA* strains were incubated for 1 hr with PBS (mock), complement-containing serum (serum), or heat-inactivated serum (HI-serum). Following treatment, the cultures were serially diluted and plated on LB agar to quantify colony-forming units (CFU), as shown on the *Y*-axis. (**C, D**) Capsule-dependent survival against T6SS assaults. T6SS-negative (Δ*hcp*) (**C**) or T6SS-positive (**D**) strains of *A. baumannii* were co-incubated with T6SS+ (white bars) or T6SS− (dashed bars) *Enterobacter cloacae*. Strains were either capsulated (WT background) or non-capsulated (Δ*itrA* background). In panel (**C**), capsulation was restored by provision of $P_{BAD}$-*itrA* on a miniTn7 transposon (Tn-*itrA*) and provision of 2% arabinose. Tn is shown for WT and mutant strains containing the transposon without a specific cargo gene. *A. baumannii* survival was quantified and is shown on the *Y*-axis. The data represent means from three independent experiments with individual values shown by the circles (± SD, indicated by error bars). Statistical significance was assessed using an ordinary one-way ANOVA test. *p < 0.05, ****p < 0.0001, ns = not significant. Detection limits (dl) were noted where applicable.

The online version of this article includes the following source data and figure supplement(s) for figure 1:

**Source data 1.** PDF file containing the original gels for *Figure 1A*, indicating the relevant bands.

**Source data 2.** Original files for gels displayed in *Figure 1A*.

**Figure supplement 1.** *A. baumannii* strain A118 produces functional T6SS.

in a T6SS-dependent manner. These results verify that *A. baumannii* A118 indeed possesses an active antibacterial T6SS when tested under standard laboratory conditions. Collectively, our findings illustrate that both the capsule and the T6SS play pivotal roles in *A. baumannii*'s defense against T6SS-mediated attacks.

## CPS-deficient *A. baumannii* exhibits increased T6SS activity

The findings described above pose an intriguing question: How does *A. baumannii*'s CPS shield the bacterium from T6SS assaults by other microbes, yet still allow it to deploy its own T6SS weaponry? Or essentially, does the capsule function as a one-way barrier? To start addressing this question, we conducted a killing experiment with *A. baumannii* A118 strains, both CPS-positive (WT) and CPS-negative (Δ*itrA*), acting as attackers. Notably, in *A. baumannii*, the CPS and protein *O*-glycosylation pathway share common enzymes, including the initiating (glycosyl)transferase ItrA, which attaches the first monosaccharide to the undecaprenol-phosphate (Und-P) lipid carrier (*Lees-Miller et al., 2013*). To distinguish between these pathways, both affected in the *itrA* mutant, we also included an *O*-glycosylation-specific mutant (Δ*pglL*) as an attacker. Initially, at a standard attacker:prey ratio of 1:1, no significant differences in prey survival were observed between CPS-positive (WT), CPS-negative (Δ*itrA*), and *O*-glycosylation (Δ*pglL*) mutant strains, with survival rates at or below the detection limit (*Figure 2A*). To enhance the sensitivity of the assay and increase its dynamic range, we adjusted the attacker:prey ratio to 1:5, reflecting better a situation where a bacterium invades an already existing community. Under these conditions, we noted decreased T6SS-mediated killing activity in the WT strain compared to the Δ*itrA* strain, while no difference was noted between the WT and Δ*pglL* strains (*Figure 2A*). This suggests that T6SS-mediated killing activity is elevated in the absence of CPS and that the *O*-glycosylation pathway does not contribute to this phenotype.

To verify the increased T6SS activity in the absence of CPS, we performed an Hcp secretion assay. This assay, a benchmark for assessing T6SS secretion functionality (*Pukatzki et al., 2006*), relies on immunodetecting the Hcp protein in the supernatant, with intracellular Hcp serving as a control. Importantly, while all strains produced Hcp at comparable levels, the amount of Hcp detected in the supernatant was considerably higher in the Δ*itrA* strain compared to the WT or the Δ*pglL* mutant (*Figure 2B*). This finding reinforces the notion that T6SS secretion activity is enhanced in the absence of the CPS. As an internal BSA precipitation control ensured that the observed differences in Hcp recovery were not due to variations in precipitation efficiency, we hypothesized that the CPS might directly impact the assembly of and/or firing by the T6SS machinery. We therefore compared T6SS structures within cells using a functional translational fusion between TssB and msfGFP (*Figure 1—figure supplement 1B*), as previously reported (*Lin et al., 2022*). To objectively assess T6SS assembly, we developed a tool designed for the automatic analysis T6SS structures in cells over a 5-min time interval (*Video 1*). Our observations revealed highly dynamic T6SS structures in nearly all WT (96.2% ± 2.8) and *itrA* mutant cells (98.0% ± 2.2) (*Figure 2C, D*). This data indicates that the capsule's presence or absence does not affect the production or assembly of the T6SS in *A. baumannii* A118.

Collectively, our findings indicate that the capsule modulates T6SS activity, as shown by the variations in killing efficiency and Hcp secretion between encapsulated and non-encapsulated strains. This suggests that the capsule may serve as an additional barrier the T6SS has to traverse to be expelled from the cell. Supporting this theory, our analysis reveals that the enhanced T6SS activity in the non-capsulated mutant (Δ*itrA*) is not due to a higher number of T6SS assemblies but likely due to an increase in the number of successful T6SS firing events. This finding is in line with previous reports on *Campylobacter jejuni*, where the T6SS was only cytotoxic to red blood cells in a capsule-deficient context (*Bleumink-Pluym et al., 2013*), leading to the hypothesis that the capsule acts as a physical barrier, limiting T6SS's ability to directly interact with target cells. Variations in capsule production have been observed in *A. baumannii*, which employs a kind of bet-hedging strategy that leads to the formation of two types of variants within the same clonal population, namely opaque and translucent colonies. These variants are capable of phenotypically switching between these states, thereby enhancing their adaptation to diverse environments (*Chin et al., 2018*). Such a strategy in capsule modulation can offer significant advantages, including protection against external threats like complement-mediated killing, as well as competitive interactions with surrounding organisms. Furthermore, the increased T6SS activity in the non-capsulated strain may provide a protective advantage against surrounding competitors in the absence of the protective capsule layer.

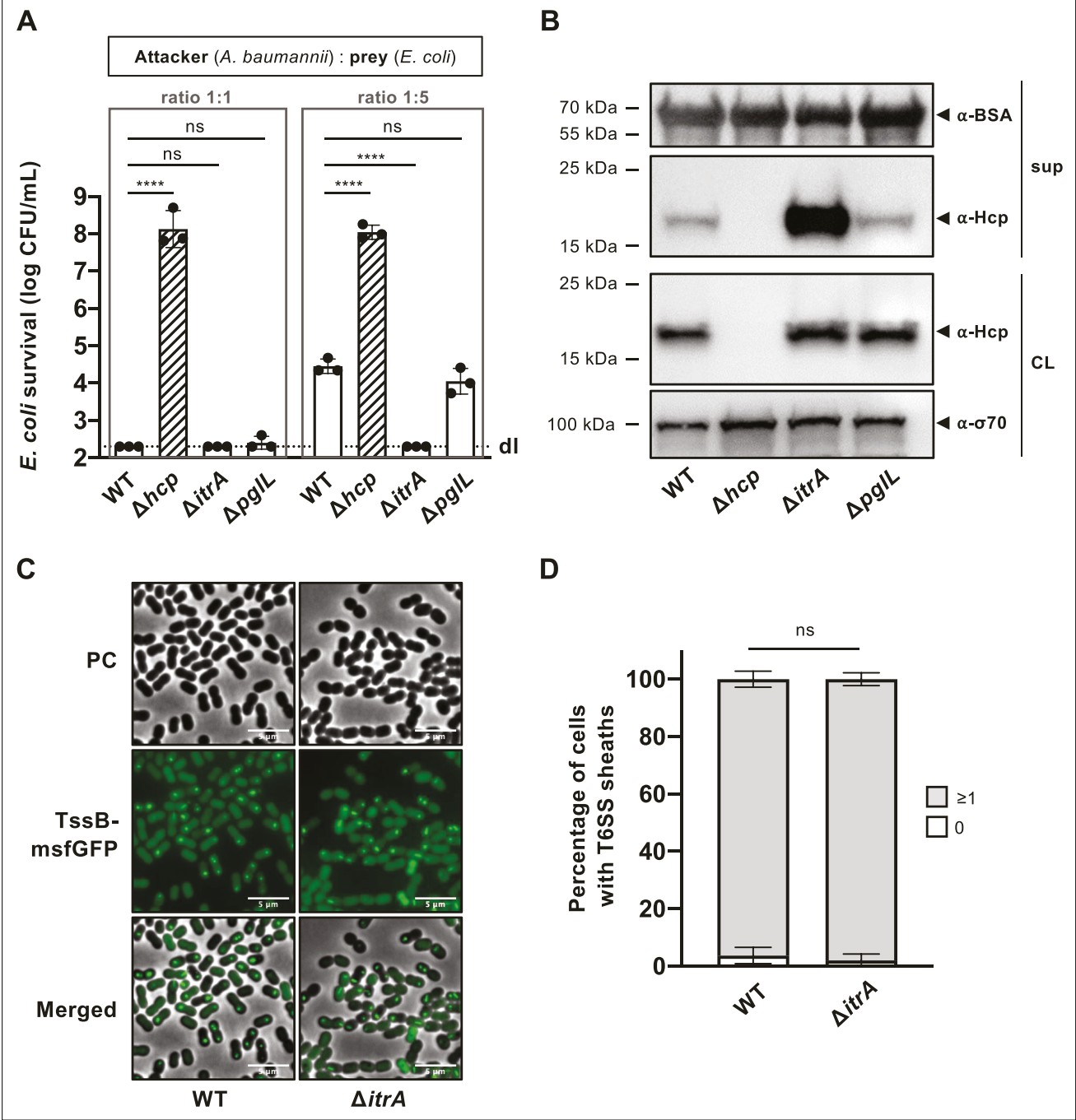

**Figure 2.** Capsular polysaccharide (CPS) interferes with T6SS activity. (**A**) Non-capsulated strains show increased T6SS killing-mediated activity. Survival of *E. coli* after encountering capsulated (wild-type, WT), T6SS-inactive (Δ*hcp*), non-capsulated (Δ*itrA*), or *O*-glycosylation (Δ*pglL*) mutant *A. baumannii*, with different attacker-to-prey ratios as indicated. Survival rates are shown as on the *Y*-axis. (**B**) Analysis of Hcp production and secretion in the strains mentioned in (**A**). Cell lysates (CL) and culture supernatants (sup) were tested through immunoblotting, using antibodies against Hcp (α-Hcp). The loading control (α-σ70) confirms equal amounts of the CL. BSA was added to supernatants and detected with α-BSA antibodies as a precipitation control. The data is representative of three independent experiments. (**C**) Fluorescence light micrographs of exponentially grown *A. baumannii* cells, either producing (WT) or not producing (Δ*itrA*) CPS, with a translational fusion (msfGFP) to the T6SS sheath protein TssB. Images include phase contrast (PC), green fluorescence (TssB-msfGFP), and a merged view of both channels. Scale bar: 5 μm. (**D**) Quantification of T6SS assembly over 5-min time-lapses in TssB-msfGFP-carrying bacteria, comparing capsulated (WT; n = 2832 cells) and non-capsulated (Δ*itrA*; n = 2831 cells) cells. The *Y*-axis shows the percentage of cells producing T6SS structures, with cells not producing T6SS in white and those producing at least one structure in gray. Data are averages from three experiments (± SD, as defined by error bars). Statistical significance compared to WT is marked, determined via an ordinary one-way ANOVA test (**A**) or a two-way ANOVA test (**D**), with ****p < 0.0001, ns = not significant. Detection limits (dl) are indicated.

*Figure 2 continued on next page*

*Figure 2 continued*

The online version of this article includes the following source data for figure 2:

**Source data 1.** PDF file containing original western blots for *Figure 2B*, indicating the relevant bands and treatments.

**Source data 2.** Original files for western blot analysis displayed in *Figure 2B*.

## Alterations in the organization of capsule material disrupt the secretion process

Having established that CPS interferes with the T6SS secretion process, we next explored whether enhancing CPS production could entirely block T6SS activity. Previous research has identified two genetic alterations that increase CPS secretion and/or production (*Geisinger and Isberg, 2015*). Indeed, Geisinger and Isberg demonstrated that a substitution in the Walker A motif of the Wzc protein, which controls the size of exported polysaccharides, induces in a mucoviscous phenotype characterized by abnormally high molecular weight polysaccharides predominantly found in the supernatant and only loosely attached to the cell. The second mechanism for elevated CPS production involves the two-component system BfmRS, recognized for its role in various cellular processes including biofilm formation, serum resistance, antibiotic resistance, and envelope stress response (*Geisinger et al., 2018*; *Russo et al., 2016*; *Tomaras et al., 2008*). The BfmS histidine kinase within this system typically represses K locus expression by phosphorylation of the response regulator BfmR (*Palethorpe et al., 2022*). Consequently, removing *bfmS* disrupts this phosphorylation cascade, resulting in the overproduction of the K locus gene cluster (*Geisinger and Isberg, 2015*).

To enhance CPS production in *A. baumannii* A118, we therefore introduced a point mutation in *wzc* encoding the Walker A motif variant [K547Q], and we also created a deletion mutant of *bfmS* (*Figure 3A*). Both modifications led to the formation of mucoviscous colonies (*Figure 3—figure supplement 1A, B*), with a noticeable difference: stretching of the Wzc[K547Q] colonies produced a string (>5 mm), a phenomenon not observed in the mucoviscous Δ*bfmS* mutant. This suggests that the capsular characteristics differ between the two mutants. Indeed, further analysis, including CPS extraction followed by Alcian blue staining and the serum-mediated killing assay, revealed distinct outcomes for these mutants. The Δ*bfmS* mutant showed increased resistance to serum-mediated killing (*Figure 3A*), aligned with an augmented presence of cell-associated CPS material (*Figure 3B*). Conversely, the Wzc[K547Q] mutant displayed heightened susceptibility to the rabbit serum (*Figure 3A*), which correlates with the faint CPS signal detected in the cell fraction by Alcian blue staining (*Figure 3B*). This indicates that dysregulation in polysaccharide chain length can adversely affect the capsule's protective properties.

To assess the impact of these altered CPS profiles on T6SS-mediated killing activity, we utilized these mutants as attackers in a killing assay. The results reveal that the Δ*bfmS* mutant exhibits a significant reduction in its ability to kill *E. coli* prey cells (*Figure 3C*). By complementing the Δ*bfmS* mutant in cis with a copy of *bfmS* under its native promoter, T6SS-mediated killing was restored to levels similar to those of the WT strain (*Figure 3—figure supplement 1C*). To rule out the possibility of the *bfmS* mutation having broad effects on T6SS production or function, we also evaluated the double mutant Δ*bfmS*Δ*itrA*, which reinstated the strain's killing ability (*Figure 3C*). These findings were consistent with the Hcp secretion profiles of these mutants. Specifically, the mutant lacking *bfmS* showed a significant impairment in its Hcp secretion activity, whereas the Δ*bfmS*Δ*itrA* double mutant reflected the secretion pattern of the Δ*itrA* strain (*Figure 3D*). This suggests that the observed phenotype indeed results from the increased production of the capsule.

To investigate whether capsule-secreted material could interact with proteins in the supernatant, we co-cultured the WT and a secretion-impaired

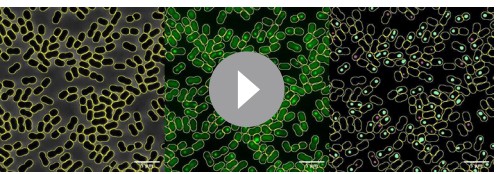

**Video 1.** Movie demonstrating an example of the image analysis pipeline. A representative movie is shown, with snapshots captured every 30 s over a 5-min duration. The panel displays split views: the phase-contrast channel on the left, the bleach- and drift-compensated fluorescence channel in the center, and the color-coded channel showing contracted and extended sheath structures on the right. A mask representing the segmented bacteria is overlaid on all three panels.

https://elifesciences.org/articles/101032/figures#video1

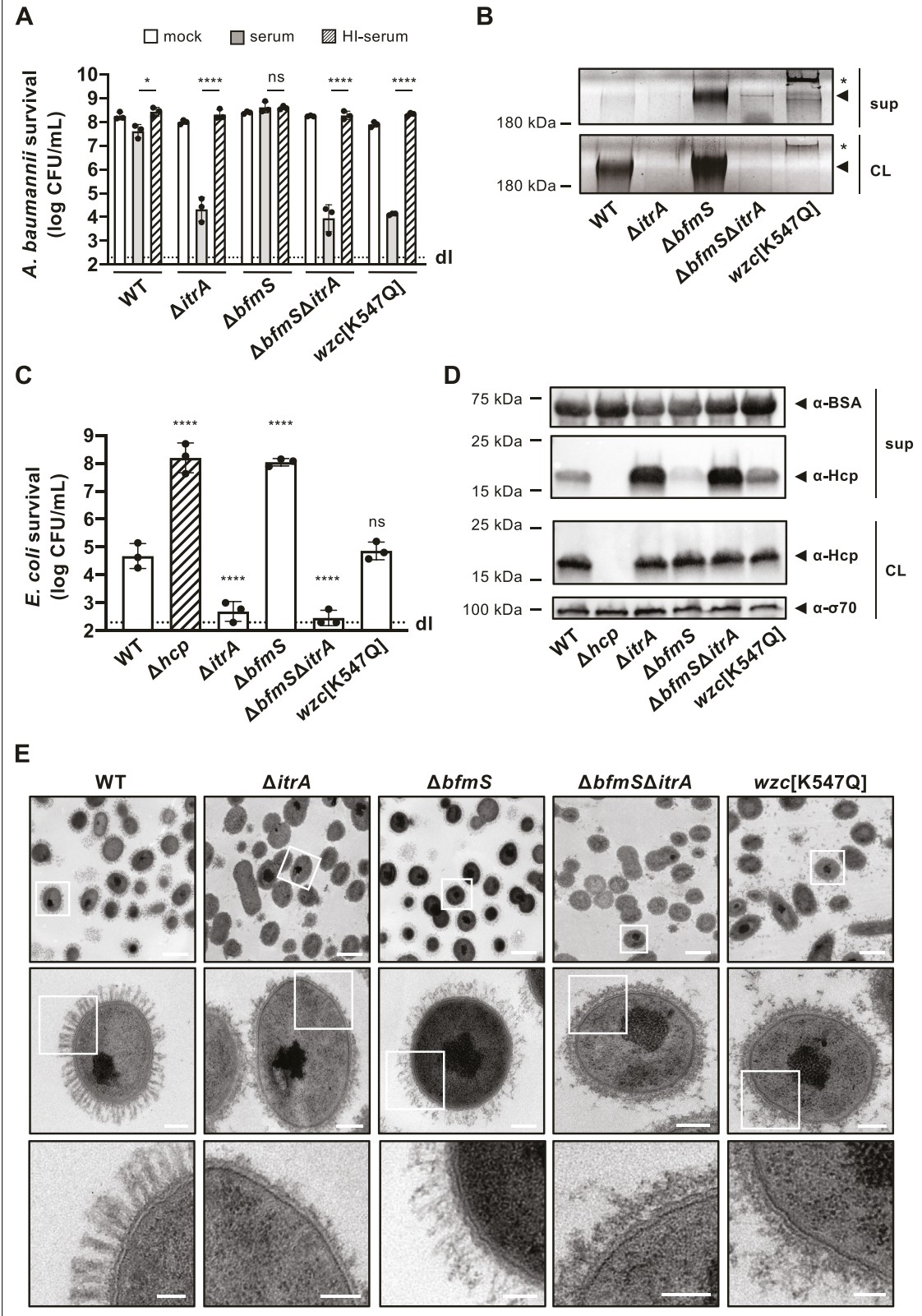

**Figure 3.** Increased capsular polysaccharide (CPS) production inhibits T6SS activity. (**A**) Complement resistance assay across *A. baumannii* strains. The assay tested the resistance against complement-containing serum of these strains: capsulated wild-type (WT), non-capsulated (Δ*itrA*), capsule-overproducing (Δ*bfmS*), a Δ*bfmS*Δ*itrA* double mutant, and a strain carrying mutated *wzc* (encoding Wzc[K547Q]). Details as described for *Figure 1B*. (**B**) Polysaccharide analysis in WT and variants described in panel A. Polysaccharides from cell lysate (CL) or supernatants (sup) were separated by

*Figure 3 continued on next page*

*Figure 3 continued*

SDS–PAGE and stained with Alcian blue. Arrowheads point to polysaccharide bands with the asterisks marking high molecular size polysaccharides. (**C**) Survival of *E. coli* prey after interaction with the *A. baumannii* strains described in panel (**A**) as attackers. A T6SS-inactive strain (Δ*hcp*, dashed bar) was added as control. The attacker-to-prey ratio of 1:5 was used. Survival rates are indicated on the *Y*-axis. Details as for *Figure 2A*. (**D**) Hcp production and secretion levels of WT and mutant *A. baumannii* strains described in panel (**A**). Details as in *Figure 2B*. (**E**) Transmission electron microscopy images of WT, Δ*itrA*, Δ*bfmS*, Δ*bfmS*Δ*itrA*, and *wzc*[K547Q] strains. White squares indicate zoomed areas. Scale bars correspond to 1, 0.2, and 0.1 µm for the top, middle, and bottom images, respectively. Data for panels (**B**), (**D**), and (**E**) are representative of three independent experiments. For panels (**A**) and (**C**), data points are averages from three experiments (± SD, shown by error bars). Statistical significance compared to the heat-inactive serum treatment (**A**) or to the WT strain (**C**) is noted above the charts, determined with an ordinary one-way ANOVA test. *p < 0.05, ****p < 0.0001, ns = not significant. Detection limits (dl) were noted where applicable.

The online version of this article includes the following source data and figure supplement(s) for figure 3:

**Source data 1.** PDF file containing the original gels for *Figure 3B*, indicating the relevant bands.

**Source data 2.** Original files for gel displayed in *Figure 3B*.

**Source data 3.** PDF file containing original western blots for *Figure 3D*, indicating the relevant bands and treatments.

**Source data 4.** Original files for western blot analysis displayed in *Figure 3D*.

**Figure supplement 1.** Deletion of *bfmS* and its effect on T6SS activity in *A. baumannii*.

**Figure supplement 1—source data 1.** PDF file containing the original gels for *Figure 3—figure supplement 1D*, indicating the relevant bands.

**Figure supplement 1—source data 2.** Original files for gel displayed in *Figure 3—figure supplement 1D*.

**Figure supplement 1—source data 3.** PDF file containing original western blots for *Figure 3—figure supplement 1E*, indicating the relevant bands and treatments.

**Figure supplement 1—source data 4.** Original files for western blot analysis displayed in *Figure 3—figure supplement 1E*.

Δ*tssB* mutant and compared their Hcp secretion profiles to the WT co-cultured with Δ*bfmS*. Notably, all strains except the Δ*bfmS* mutant secreted comparable amounts of polysaccharide into the supernatant (*Figure 3—figure supplement 1D*). We observed no significant differences in the levels of Hcp protein secretion between the two co-culture conditions (*Figure 3—figure supplement 1E*), indicating that the secretion defect seen in Δ*bfmS* is attributable to an impairment in secretion rather than to interactions of Hcp with CPS in the supernatant. We also engineered a *bfmS* deletion in two environmental *A. baumannii* isolates (29D2 and 86II/2C) (*Wilharm et al., 2017*). As illustrated in *Figure 3—figure supplement 1F*, both strains are capable of producing an antibacterial T6SS. Importantly, the deletion of *bfmS* in these strains also resulted in the impairment of the T6SS-mediated killing. These findings indicate that *bfmS* affects T6SS-mediated killing activity across different strains.

Unlike the Δ*bfmS* phenotypes, the Wzc[K547Q] variant demonstrated T6SS-mediated killing similar to that observed in the WT, as shown in *Figure 3C*. Consistently, this variant also exhibited Hcp secretion levels that appeared comparable (or even increased) to those of the WT (*Figure 3D*). This finding is in line with the observation that the Wzc[K547Q] variant was not protected from complement-mediated killing (*Figure 3A*) and produces minimal surface-bound CPS (*Figure 3B*).

To gain more insight into the ultrastructure of the capsule in the different genetic backgrounds, we imaged cells using transmission electron microscopy (TEM) (*Figure 3E*). The WT cells were surrounded by material forming large finger-like projections extending about 150 nm from the cell surface, arranged in a semi-regular pattern of projections and spaces. As expected, these structures were absent in the Δ*itrA* mutant, confirming its essential role in capsule assembly. Notably, the Wzc[K547Q] variant also lacked these structures, appearing similar to the Δ*itrA* mutant. However, we observed a significant presence of what is presumed to be capsular material floating in the medium surrounding the cells, with additional material potentially being lost during the fixation process (*Figure 3E*). This detached CPS aligns with the Alcian blue staining results (*Figure 3B*) and could explain the observed differences in the string test results for the Wzc[K547Q] colonies compared to the Δ*bfmS* mutant (*Figure 3—figure supplement 1A, B*). The creation of such a viscous environment by the release of long-chain CPS may therefore impact T6SS activity, explaining the decreased killing ability compared to the Δ*itrA* mutant (*Figure 3C*). In contrast, the *bfmS* mutant exhibited a dense, tangled, mesh-like network of CPS covering the cell surface, similar to the WT but without the clear periodic spaces. As expected, this dense capsule network was absent in the Δ*bfmS*Δ*itrA* double deletion mutant (*Figure 3E*).

Taken all together, these results indicate that disruption in the organized, finger-like structure of the capsule, as seen with overexpression of the K-locus, leads to a suppression of T6SS-mediated killing activity and blocks Hcp secretion. This observation highlights the importance of CPS's surface organization in affecting the extracellular secretion process. It is tempting to speculate that, within the WT scenario, T6SS may deploy through gaps akin to arrow-slit in the capsule's mesh, a process that becomes unfeasible when CPS organization is disrupted. This concept mirrors a hypothesis suggested by Toska et al. for *Vibrio cholerae*, where T6SS secretes through biofilm-associated EPS (*Toska et al., 2018*). An alternative explanation might be that capsule overexpression enhances polysaccharide dispersion into the surroundings. Coupled with changes to the capsule directly attached to the cell surface, this could effectively increase the spatial gap between cells, impeding T6SS functionality.

## Antibiotics-induced CPS production impairs T6SS activity

It has been shown that the *A. baumannii* isolate 17978 boosts CPS production via the BfmRS two component system in response to sub-minimal inhibitory concentrations (sub-MIC) of chloramphenicol (*Geisinger and Isberg, 2015*). When we exposed *A. baumannii* A118 to various chloramphenicol concentrations, we found that the capsule induction by the antibiotic was dose-dependent, as evidenced by increased CPS presence in the supernatant (*Figure 4A*). We next asked whether T6SS activity inhibition seen in the ΔbfmS mutant could also be induced under antibiotic-triggered capsule overproduction conditions in the WT background. Unfortunately, the Hcp secretion assay did not yield conclusive results due to contamination from cytoplasmic material, indicating that chloramphenicol exposure caused partial cell lysis (LB+Cm), which was not observed in the untreated control (LB) (*Figure 4B*). However, we noticed enhanced T6SS-mediated killing in the non-capsulated strain (ΔitrA) compared to the capsulated WT under antibiotic exposure, with no difference in T6SS-mediated killing between the ΔbfmS control strain and the antibiotic-treated WT (*Figure 4C*). This suggests that chloramphenicol-induced capsule production disrupts T6SS-mediated killing activity. It is important to note that antibiotic treatment alters *A. baumannii*'s growth, which may change the attacker:prey ratio during the assay and affect the experimental results. However, this effect should similarly influence the ΔitrA mutant, which maintains effective killing even in the presence of chloramphenicol. To improve the assay's dynamic range, we adjusted the attacker:prey ratio to 5:1 (*Figure 4C*). Under these conditions, we still observed increased T6SS-mediated killing by the non-capsulated strain (ΔitrA) compared to the capsulated WT under antibiotic exposure, supporting the conclusion that chloramphenicol exposure inhibits T6SS-mediated killing due to increased capsule production.

Collectively, this data suggests that the inhibition of T6SS by increased capsule production, as observed with the ΔbfmS mutant, could be relevant in natural conditions that *A. baumannii* might encounter, such as the presence of sub-MIC antibiotics in the environment. Indeed, this mucoid state has been observed with other antibiotics apart from chloramphenicol, some of which are used in clinical settings (*Geisinger and Isberg, 2015*; *Traub and Bauer, 2000*). Interestingly, Geisinger and Isberg demonstrated that the antibiotic-induced enhancement of capsule production represents a non-mutational phenotype, which can be reversed upon removal of the antibiotic (*Geisinger and Isberg, 2015*). It is therefore tempting to speculate that this inverse relationship between capsule production and T6SS activity may provide adaptive advantages in response to environmental changes and competitive interactions with other bacteria, as proposed by *Weber et al., 2015*.

## Capsule-overproducing strains assemble fewer T6SS but retain sensory function

A recent investigation into *A. baylyi* revealed the presence of TslA, a periplasmic protein essential for precise assembly of the T6SS machinery at points of contact with other cells, aiming to prevent wasteful T6SS firing events (*Lin et al., 2022*). Importantly, the periplasmic *Acinetobacter* type six secretion system-associated A protein (AsaA), which is the TslA homolog in *A. baumannii*, has been shown to play a role in efficient T6SS activity (*Li et al., 2019*). Indeed, Li et al. suggested that AsaA/TslA impacts the assembly or stability of the T6SS through its interaction with the membrane complex protein TssM (*Li et al., 2019*). These findings led to the speculation that in the ΔbfmS mutant, the elevated levels of capsule production could obstruct the environmental sensing function of TslA, thereby reducing the likelihood of T6SS assembly.

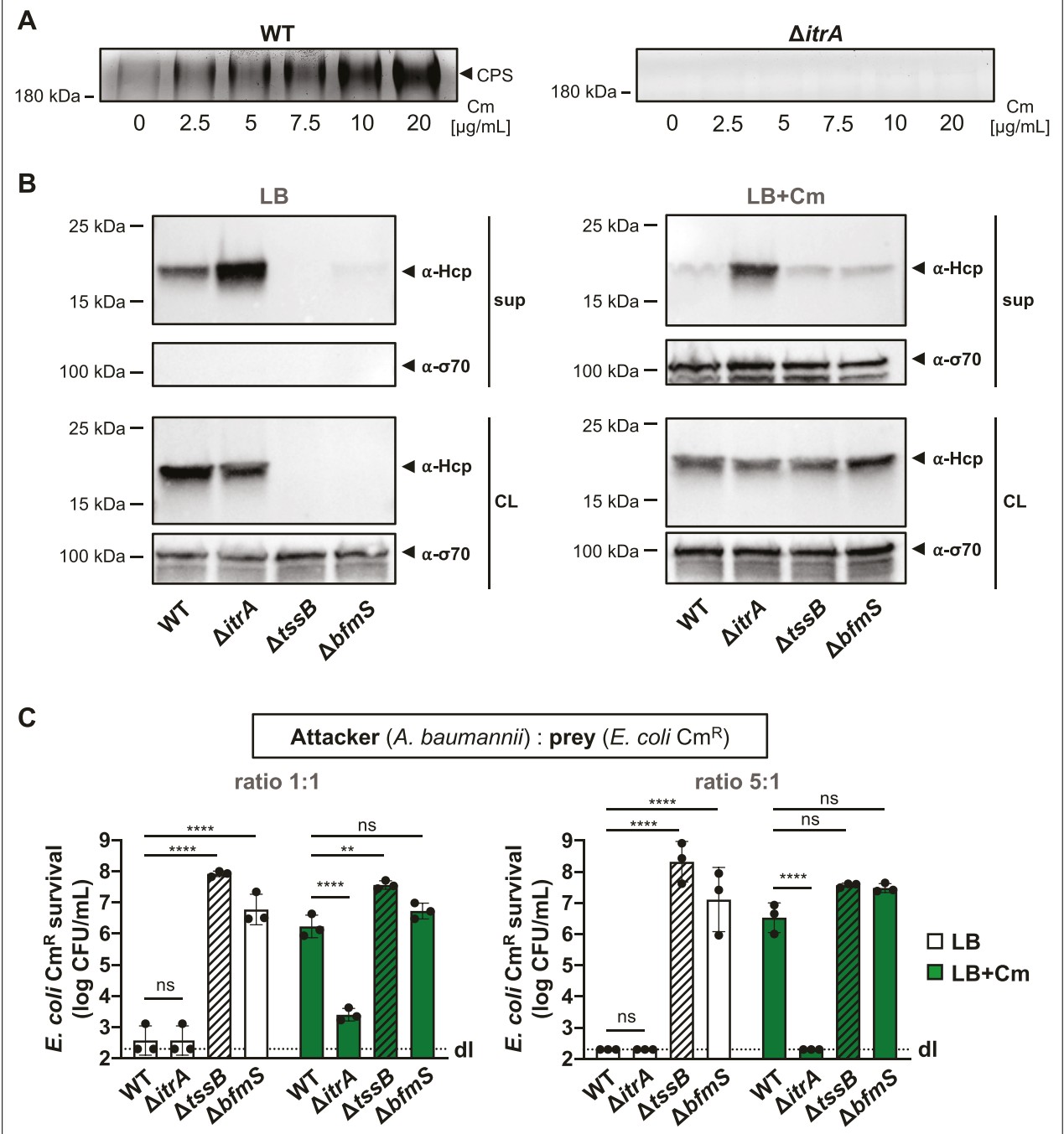

**Figure 4.** T6SS inhibition upon capsular polysaccharide (CPS) overproduction due to antibiotic treatment. (**A**) Capsule production in wild-type (WT, left panel) or capsule-deficient (ΔitrA, right panel) strains was induced using varying concentrations of chloramphenicol, as indicated. Polysaccharides from supernatant were precipitated, separated by SDS–PAGE, and visualized with Alcian blue staining. The arrowhead marks the polysaccharide band. (**B**) Hcp secretion in chloramphenicol-untreated (LB) or treated (LB+Cm) cells. Hcp production and secretion were analyzed in WT, ΔitrA, ΔtssB, and ΔbfmS strains through immunoblotting. Details as described in *Figure 2B*. σ-70 detection served as loading (CL) and lysis control (sup). (**C**) Survival of chloramphenicol-resistant (Cm^R) *E. coli* after contact with various *A. baumannii* attackers: capsulated (WT), non-capsulated (ΔitrA), T6SS-inactive (ΔtssB, dashed bars), or capsule-overexpressing (ΔbfmS) strains. Assay was conducted at different attacker-to-prey ratios, as indicated, under two conditions: unexposed (white bars) and exposed to 20 µg/ml chloramphenicol (green bars) to induce capsule production. Survival rates are indicated on the *Y*-axis. For panel (**C**), data points are averages from three independent experiments (± SD, shown by error bars). Statistical significance was determined with an ordinary one-way ANOVA test. **$p < 0.01$, ****$p < 0.0001$, ns = not significant. Detection limits (dl) were noted where applicable.

The online version of this article includes the following source data for figure 4:

**Source data 1.** PDF file containing the original gels for *Figure 4A*, indicating the relevant bands.

*Figure 4 continued on next page*

*Figure 4 continued*

**Source data 2.** Original files for gel displayed in *Figure 4A*.

**Source data 3.** PDF file containing original western blots for *Figure 4B*, indicating the relevant bands and treatments.

**Source data 4.** Original files for western blot analysis displayed in *Figure 4B*.

To test this hypothesis, we analyzed the dynamics of the TssB-msfGFP fusion in WT, Δ*tslA*, and Δ*bfmS* backgrounds using time-lapse microscopy over a 5-min time span (*Figure 5A, B*). We noted a significant 5.6-fold decrease in T6SS assembly in the Δ*tslA* mutant (17.6 ± 8.7%) compared to the WT (99.0 ± 1.4%). Meanwhile, the Δ*bfmS* mutant exhibited a more moderate 1.9-fold reduction (50.8 ± 8.8%) in comparison to the WT. Remarkably, the Δ*bfmS*Δ*itrA* double mutant showed T6SS assembly rates (97.6 ± 1.0%) similar to both the WT and Δ*itrA* (99.2 ± 1.0%) strains. These findings suggest that the capsule's overproduction in Δ*bfmS* only partially influences T6SS assembly. Of note, the decreased fluorescence intensity observed in the Δ*bfmS* and Δ*bfmS*Δ*itrA* strains compared to WT, Δ*itrA*, and Δ*tslA* strains (*Figure 5A*; *Figure 5—figure supplement 1A*) remains unexplained and may be related to a physiological change in the Δ*bfmS* background and potential folding issues of GFP under those conditions. Nonetheless, western blot analysis confirmed consistent fusion protein levels across *bfmS*-positive and -negative strains (*Figure 5—figure supplement 1B*), indicating that TssB-msfGFP production is unaffected in these mutants.

To further investigate how the absence of contact sensing affects T6SS activity, we delved into the role of TslA in *A. baumannii* A118 by conducting a killing experiment (*Figure 5C*) and a Hcp secretion assay (*Figure 5D*). In line with results obtained previously in *A. baylyi* (*Ringel et al., 2017*) and *A. baumannii* ATCC17978 (*Kandolo et al., 2023*; *Li et al., 2019*), the removal of *tslA* led to a decrease in T6SS-mediated killing and the amount of secreted Hcp. Interestingly, the *tslA* mutant displayed significantly higher levels of killing (*Figure 5C*) and Hcp secretion (*Figure 5D*) compared to the Δ*bfmS* mutant under identical experimental conditions.

Taken together, these findings collectively indicate that the diminished killing and secretion performance seen in the Δ*bfmS* mutant cannot be solely attributed to a defect in cell–cell contact sensing and T6SS assembly. Indeed, despite assembling around three times more T6SS structures compared to the Δ*tslA* mutant, the Δ*bfmS* mutant exhibits a T6SS killing activity that is 100 times less effective. This evidence points to the conclusion that T6SS assembly is not the limiting factor for the T6SS inhibition in the Δ*bfmS* mutant.

## Prolonged secretion inhibition triggers Hcp degradation

Given that our data indicate an inhibition of T6SS activity by CPS upregulation, we explored the possibility of regulatory cross-talk between these two processes during prolonged capsule overproduction. To address this question, we assessed Hcp production levels in various capsule mutant backgrounds using strains in stationary phase (15–16 hr of growth) (*Figure 6A*). Our observations revealed that, in stationary phase cultures, Hcp was undetectable in the Δ*bfmS* mutant but present at WT levels in the Δ*bfmS*Δ*itrA* double mutant (*Figure 6A*). This contrasts with results from exponentially growing cultures, where all strains produced Hcp at comparable levels (*Figure 3D*). Given the impaired Hcp secretion in the Δ*bfmS* mutant, we hypothesized that intracellular accumulation of Hcp or another unidentified signal may trigger a feedback mechanism that downregulates Hcp production.

Interestingly, a recent study reported that *V. cholerae* can sense Hcp levels and regulate T6SS expression accordingly (*Manera et al., 2021*). The authors demonstrated that the RpoN-dependent regulator VasH interacts with Hcp, influencing the expression of auxiliary T6SS clusters that include the *hcp* genes. However, unlike *V. cholerae*, the main T6SS cluster in *A. baumannii* does not contain genes for bacterial enhancer-binding proteins (bEBP) such as VasH (*Figure 1—figure supplement 1A*), indicating a potentially different regulatory mechanism. To further explore the effects of Hcp accumulation in *A. baumannii*, we employed the secretion-deficient Δ*tssB* mutant. We measured Hcp levels under two different growth conditions: exponential and stationary phases (*Figure 6B*). Surprisingly, similar to the Δ*bfmS* mutant, deletion of *tssB* led to the disappearance of Hcp in stationary phase. Comparable results were obtained with secretion-impaired mutants of *A. baumannii* strains 29D2 and 86II/2C (*Figure 6—figure supplement 1A*), suggesting that Hcp downregulation upon blocking secretion might be a common feature in *A. baumannii*. These findings suggest that the reduced T6SS

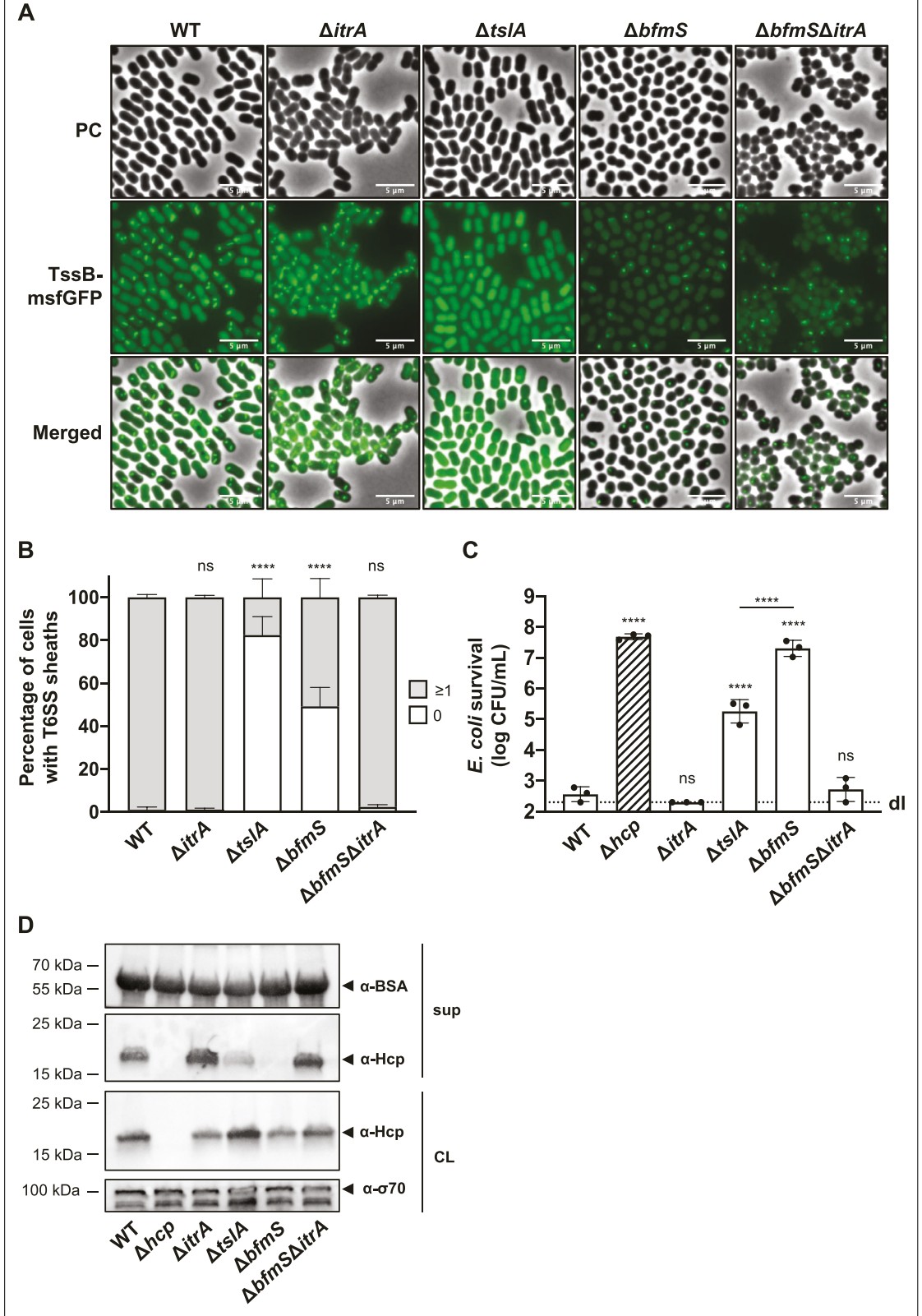

**Figure 5.** T6SS inhibition in capsular polysaccharide (CPS) overproducing strain goes beyond inability of cell-to-cell contact sensing. (**A**) Fluorescence light micrographs of TssB-msfGFP-producing *A. baumannii*. Strain backgrounds: capsulated (wild-type, WT), non-capsulated (Δ*itrA*), cell contact sensing mutant (Δ*tslA*), capsule-overexpressing (Δ*bfmS*), and Δ*bfmS*Δ*itrA* double mutant. Details as described for *Figure 2C*. Scale bar: 5 µm. (**B**) Quantification of T6SS structures in the *A. baumannii* strains described in panel (**A**). Details as for *Figure 2D*. Number of analyzed cells was 3041, 2685, 2805, 3667, and

*Figure 5 continued on next page*

Figure 5 continued

4800 for the strains indicated on the *X*-axis. Data are averages from three independent experiments (± SD, as defined by error bars). (**C**) Survival rates of *E. coli* prey after exposure to the *A. baumannii* WT, Δ*itrA*, Δ*tslA*, Δ*bfmS*, and Δ*bfmS* attacker strains with native (non-fused) *tssB*. An attacker-to-prey ratio of 1:1 was used. Survival is indicated on the *Y*-axis. Bars indicate mean values (± SD, as shown by error bars). (**D**) Hcp production and secretion were analyzed for the same *A. baumannii* strains as in panel (**C**). Experimental details as for *Figure 2B*. Statistical analyses show the significance compared to WT conditions, utilizing a two-way ANOVA test for (**B**) and an ordinary one-way ANOVA test for (**C**). ****$p < 0.0001$, ns = not significant. Detection limits (dl) are indicated.

The online version of this article includes the following source data and figure supplement(s) for figure 5:

**Source data 1.** PDF file containing original western blots for *Figure 5D*, indicating the relevant bands and treatments.

**Source data 2.** Original files for western blot analysis displayed in *Figure 5D*.

**Figure supplement 1.** TssB-msfGFP-producing *A. baumannii* strains.

**Figure supplement 1—source data 1.** PDF file containing original western blots for *Figure 5—figure supplement 1B*, indicating the relevant bands and treatments.

**Figure supplement 1—source data 2.** Original files for western blot analysis displayed in *Figure 5—figure supplement 1B*.

assembly and activity observed in the Δ*bfmS* mutant increase the cytoplasmic pool of Hcp protein, which then triggers the downregulation or degradation of Hcp.

To determine if the observed phenotype was specific to Hcp or affected other T6SS components, we assessed TssB-GFP levels using GFP antibodies in the Δ*bfmS* strain (*Figure 6—figure supplement 1B*). We observed no significant reduction in TssB-GFP levels in the Δ*bfmS* strain compared to other strains carrying *tssB-msfgfp*, suggesting that the downregulation or degradation might be specific to Hcp. To further understand the timing of this phenotype, we monitored Hcp protein production (*Figure 6C*) and *hcp* mRNA levels (*Figure 6D*) in both the WT and the secretion-impaired Δ*tssB* mutant over a 16-hr period. Notably, the decrease in Hcp protein levels in Δ*tssB* occurred between 6 and 11 hr of growth (*Figure 6C*), coinciding with the transition to late stationary phase (*Figure 6—figure supplement 1C*). However, there were no statistically significant changes in *hcp* transcript levels between the WT and the Δ*tssB* mutant throughout the experiment (*Figure 6D*). These results suggest that the effects of secretion impairment and Hcp intracellular accumulation may be regulated post-transcriptionally.

To further explore this regulatory mechanism, we attempted to overexpress Hcp in both the WT and Δ*hcp* backgrounds, monitoring *hcp* mRNA levels (*Figure 7A*) and Hcp protein production (*Figure 7B*). Compared to the empty vector control, a statistically insignificant increase in *hcp* mRNA levels was overserved in the WT background (*Figure 7A*), resulting in a slight increase in Hcp protein levels as assessed by Western blotting (*Figure 7B*). In contrast, in the Δ*hcp* background containing the *hcp*-carrying plasmid (Δ*hcp* + p-*hcp*), we observed similarly high transcript levels as for the WT condition (WT + p-*hcp*) (*Figure 7A*), yet Hcp protein production remained largely undetectable (*Figure 7B*). This observation suggests that, despite careful genetic engineering, the Δ*hcp* mutant may exhibit a polar effect that leads to Hcp accumulation upon complementation in trans, thereby mimicking the secretion-impaired Δ*tssB* mutant. In contrast, Hcp accumulation is not observed in the WT, as this strain retains its secretion capability. Notably, when *hcp* was overexpressed in *E. coli* as a control condition, both the transcript and the protein were successfully produced and detected (*Figure 7— figure supplement 1A*).

The findings suggest that Hcp is regulated at the post-transcriptional level, potentially through a degradation mechanism. ClpXP and Lon proteases are known to play crucial roles in various stress responses, specifically in degrading misfolded or accumulated proteins to mitigate proteotoxic stress (*Sauer and Baker, 2011*). To investigate the potential involvement of these general proteases in the post-transcriptional regulation of Hcp, we generated individual Δ*clpXP* and Δ*lon* mutants, as well as combinations with a T6SS secretion-impaired background (Δ*tssB*Δ*clpXP* and Δ*tssB*Δ*lon*). The *lon* mutant strains (Δ*lon* and Δ*tssB*Δ*lon*) consistently showed lower Hcp levels than the WT, suggesting that Lon is not responsible for Hcp degradation in stationary phase cultures. In contrast, deletion of *clpXP* resulted in intracellular Hcp accumulation over time in both the WT and *tssB*-deficient backgrounds, indicating that ClpXP may be involved in the degradation mechanism of Hcp when secretion is impaired in the otherwise WT background (*Figure 7C*). Supporting this idea is the observation that, although undetectable when secretion is blocked in the Δ*tssB* and Δ*bfmS* strains, Hcp levels were

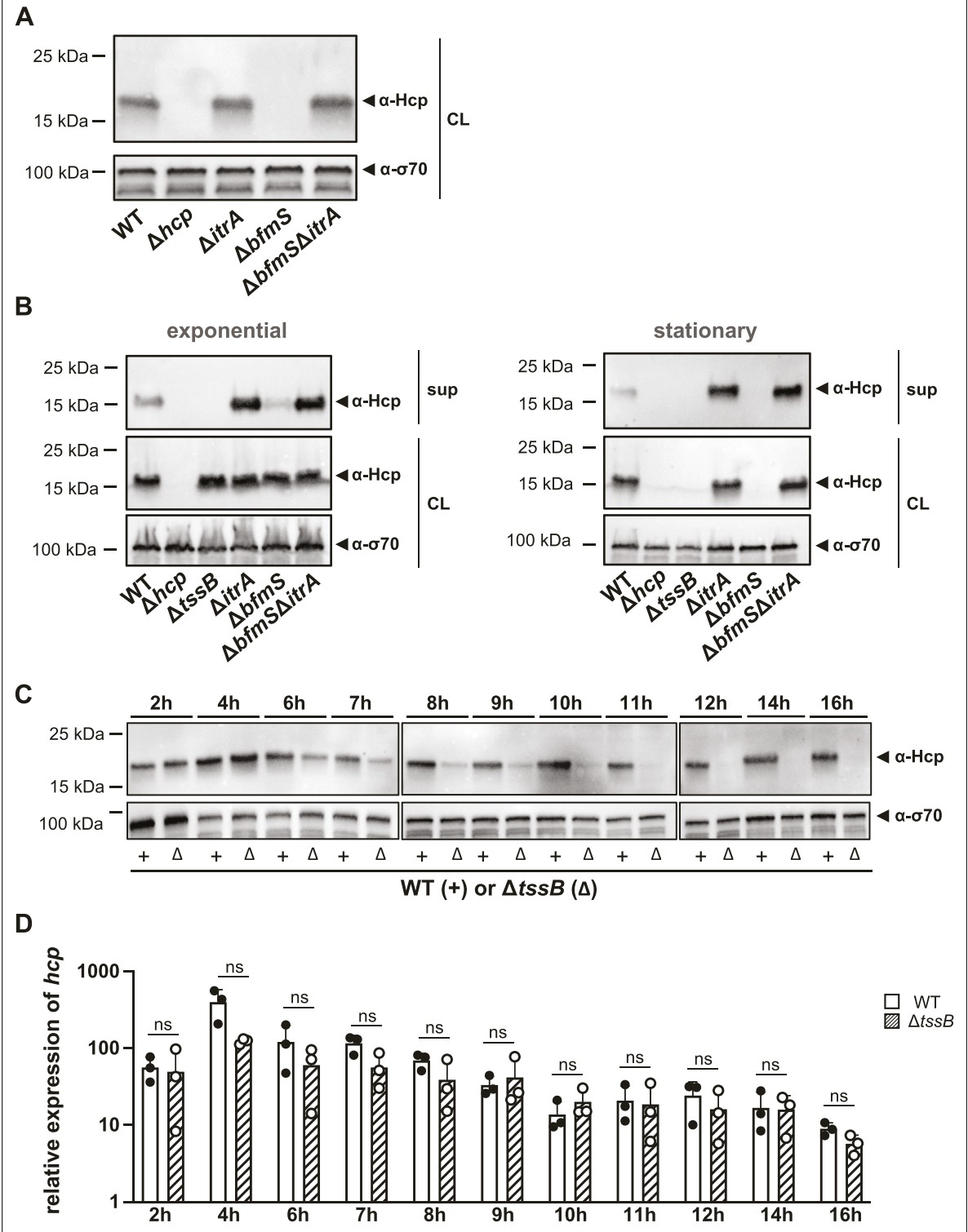

**Figure 6.** Secretion-impaired strains degrade Hcp during stationary phase. (**A**) Immunoblot analysis of Hcp protein levels in cell lysates (CL) of capsulated (wild-type, WT), T6SS-inactive (Δ*hcp*), non-capsulated (Δ*itrA*), capsule-overexpressing (Δ*bfmS*), and Δ*bfmS*Δ*itrA* mutant strains grown to stationary phase. (**B**) Comparative analysis of Hcp production and secretion in *A. baumannii* strains during exponential (left) and stationary (right) growth phases. Strains as explained in panel (**A**) with the addition of a secretion-impaired Δ*tssB* mutant. Details as described for *Figure 2B*. (**C, D**) Hcp abundance is regulated at the post-translational level. (**C**) Hcp protein production over a 16-hr period in the WT strain (white bars) versus the secretion-impaired strain (Δ*tssB*, dashed bars), as analyzed by immunoblotting. (**D**) Relative *hcp* gene expression levels over the same 16-hr period in the WT strain (white bars) versus the secretion-impaired strain (Δ*tssB*, dashed bars). These results are representative of three independent experiments, and the

*Figure 6 continued on next page*

*Figure 6 continued*

bars show the mean (± SD, as defined by error bars). Statistical analyses were performed on log-transformed data using a two-way ANOVA. ns = not significant.

The online version of this article includes the following source data and figure supplement(s) for figure 6:

**Source data 1.** PDF file containing original western blots for *Figure 6A*, indicating the relevant bands and treatments.

**Source data 2.** Original files for western blot analysis displayed in *Figure 6A*.

**Source data 3.** PDF file containing original western blots for *Figure 6B*, indicating the relevant bands and treatments.

**Source data 4.** Original files for western blot analysis displayed in *Figure 6B*.

**Source data 5.** PDF file containing original western blots for *Figure 6C*, indicating the relevant bands and treatments.

**Source data 6.** Original files for western blot analysis displayed in *Figure 6C*.

**Figure supplement 1.** Hcp degradation is conserved across *A. baumannii* strains.

**Figure supplement 1—source data 1.** PDF file containing original western blots for *Figure 6—figure supplement 1A*, indicating the relevant bands and treatments.

**Figure supplement 1—source data 2.** Original files for western blot analysis displayed in *Figure 6—figure supplement 1A*.

**Figure supplement 1—source data 3.** PDF file containing original western blots for *Figure 6—figure supplement 1B*, indicating the relevant bands and treatments.

**Figure supplement 1—source data 4.** Original files for western blot analysis displayed in *Figure 6—figure supplement 1B*.

comparable to WT in the Δ*tssB*Δ*clpXP* and Δ*bfmS*Δ*clpXP* double mutants (*Figure 7—figure supplement 1B*).

Given the time-dependent nature of this degradation, we analyzed the transcript levels of *clpX* and *clpP* at various times (2, 6, 8, and 12 hr) before, during, and after the degradation of Hcp, both under secretion-permissive (WT) and non-permissive (Δ*tssB*) conditions (*Figure 7—figure supplement 1C*). We observed an increase in *clpX* transcripts at 6 hr, corresponding with the transition into stationary phase (*Figure 6—figure supplement 1C*). However, there were no statistically significant differences in the expression levels of *clpX* and *clpP* between the WT and the Δ*tssB* background at any of the tested time points.

The ClpXP degradation system recognizes its substrates via the C-terminal region, for instance for proteins that were tagged by the SsrA system (*Sauer and Baker, 2011*), and binds it within the axial pore of the ClpX ATPase (*Martin et al., 2008*), facilitating the enzyme's ability to unfold substrates and translocate polypeptides into ClpP for degradation (*Wawrzynow et al., 1995*; *Wojtkowiak et al., 1993*). The canonical sequence of the SsrA-tag, consisting of 11 residues (AADENYNYALAA), is recognized by ClpX at the last three C-terminal amino acids (*Flynn et al., 2001*). Interestingly, when represented as a hexamer, the crystal structure of the Hcp protein from *A. baumannii* strain AB0057 (*Ruiz et al., 2015*) shows that its C-terminal domain is exposed, potentially making it accessible for interaction with ClpX and subsequent degradation (*Figure 7—figure supplement 1D*). Additionally, the last 11 C-terminal residues of the *A. baumannii* A118 Hcp protein are SLSNNTASYAA. Thus, one can speculate that when Hcp fails to be secreted and accumulates, this 'SsrA-like' tag might be recognized by the ClpXP protease machinery, leading to degradation. However, under secretion-permissive conditions, the Hcp hexamer is enclosed within the contractile sheath, thereby hiding the protein's C-terminus and protecting Hcp from degradation.

To assess the importance of the Hcp C-terminus, we constructed plasmids encoding Hcp variants lacking the last 11 amino acids (p-*hcp*[CTDΔ11]) or with Ala-to-Asp substitutions in the last two amino acids (p-*hcp*[A166D/A167D]). In *E. coli*, both variants were produced at levels similar to native Hcp (*Figure 7—figure supplement 1E*). Importantly, in the Δ*hcp* background of *A. baumannii*, we also detected high protein levels of these two modified Hcp versions (p-*hcp*[CTDΔ11] and p-*hcp*[A166D/A167D]), whereas native Hcp was again heavily degraded (p-*hcp*) (*Figure 7D*). These results suggest that the Hcp C-terminus plays a crucial for its degradation by the ClpXP machinery. More broadly, these observations indicate that *A. baumannii* has evolved a sophisticated regulation of its T6SS, closely tied to the strain's secretion capacity to prevent unnecessary protein accumulation.

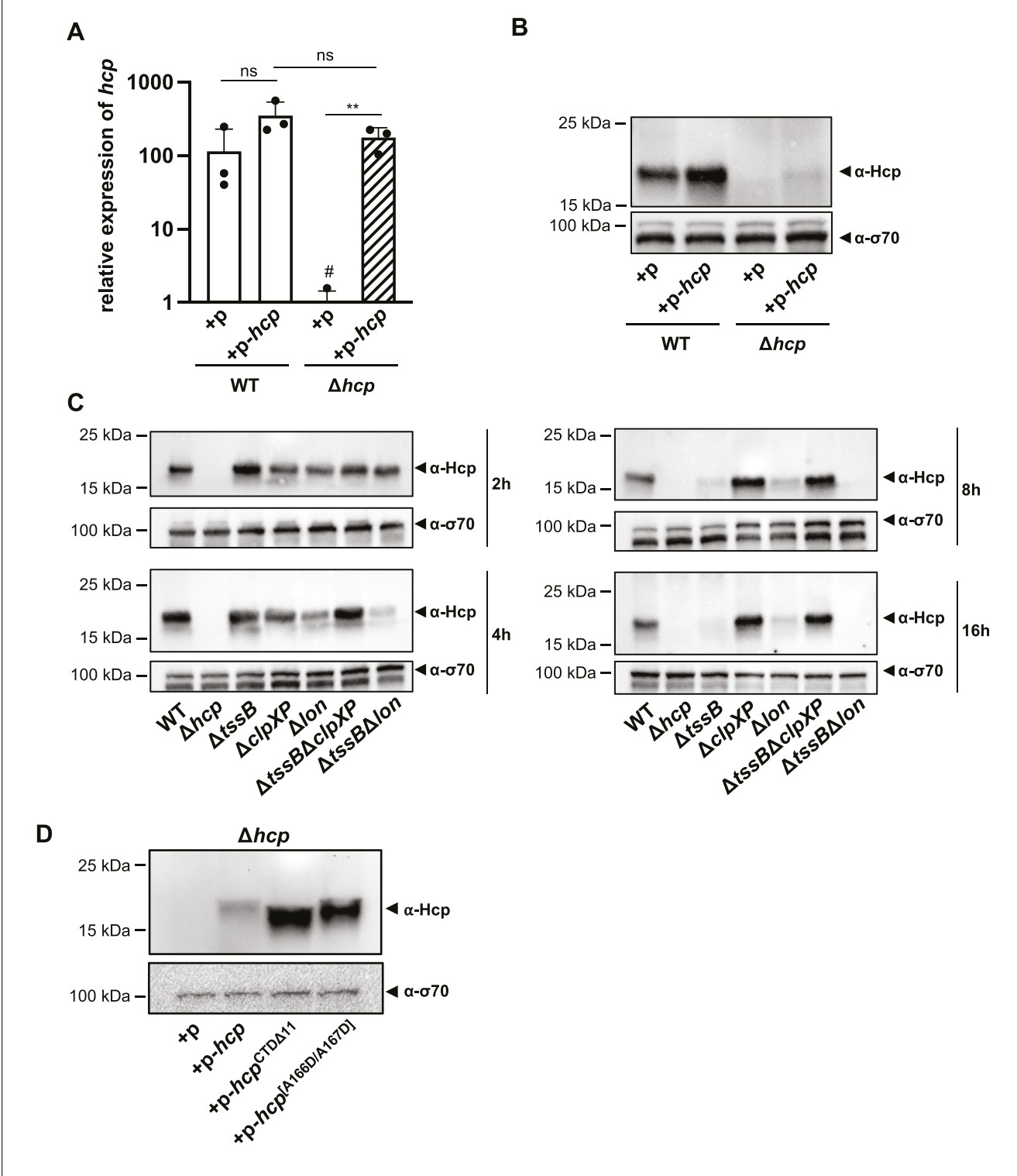

**Figure 7.** ClpXP protease machinery mediates Hcp degradation. (**A**) Relative expression levels of *hcp* in wild-type (WT) or the *hcp* mutant, either carrying an empty plasmid (+p) or a plasmid designed for *hcp* overexpression (+p *hcp*). # denotes nonspecific amplification noise. (**B**) Hcp abundance in the same strains as shown on panel A, assessed by immunoblotting. (**C**) Hcp accumulation over time in *clpXP*-deficient mutants. Hcp levels were examined in cell lysates from various *A. baumannii* strains including capsulated WT, T6SS-inactive (Δ*hcp* and Δ*tssB*), and protease-deficient (Δ*clpXP* and Δ*lon*) mutants, or strains lacking multiple genes (Δ*tssB*Δ*clpXP* and Δ*tssB*Δ*lon*). The bacteria were grown over 2-, 4-, 8-, and 16-hr growth periods. Immunoblot analyses were performed as described for *Figure 2B*. (**D**) The C-terminus of Hcp is essential for its degradation by ClpXP. Hcp abundance

*Figure 7 continued on next page*

*Figure 7 continued*

in the *hcp* mutant carrying either an empty plasmid (+p), a plasmid encoding Hcp (+p-*hcp*), a plasmid encoding a C-terminus-deficient variant of Hcp (+p-*hcp*[CTDΔ11]), or a plasmid encoding Hcp with substitutions of the last two amino acid (+p-*hcp*[A166D/A167D]). These results are representative of three independent experiments and bars show the mean (± SD, as shown by error bars). Statistical significance was assessed on log-transformed data using a two-way ANOVA. **$p < 0.01$, ns = not significant.

The online version of this article includes the following source data and figure supplement(s) for figure 7:

**Source data 1.** PDF file containing original western blots for *Figure 7B*, indicating the relevant bands and treatments.

**Source data 2.** Original files for western blot analysis displayed in *Figure 7B*.

**Source data 3.** PDF file containing original western blots for *Figure 7C*, indicating the relevant bands and treatments.

**Source data 4.** Original files for western blot analysis displayed in *Figure 7C*.

**Source data 5.** PDF file containing original western blots for *Figure 7D*, indicating the relevant bands and treatments.

**Source data 6.** Original files for western blot analysis displayed in *Figure 7D*.

**Figure supplement 1.** Hcp degradation by ClpXP.

**Figure supplement 1—source data 1.** PDF file containing original western blots for *Figure 7—figure supplement 1A*, indicating the relevant bands and treatments.

**Figure supplement 1—source data 2.** Original files for western blot analysis displayed in *Figure 7—figure supplement 1A*.

**Figure supplement 1—source data 3.** PDF file containing original western blots for *Figure 7—figure supplement 1B*, indicating the relevant bands and treatments.

**Figure supplement 1—source data 4.** Original files for western blot analysis displayed in *Figure 7—figure supplement 1B*.

**Figure supplement 1—source data 5.** PDF file containing original western blots for *Figure 7—figure supplement 1E*, indicating the relevant bands and treatments.

**Figure supplement 1—source data 6.** Original files for western blot analysis displayed in *Figure 7—figure supplement 1E*.

## Conclusion

In conclusion, our study reveals a novel role of the CPS in *A. baumannii*, highlighting its complex interaction with the T6SS, which is crucial for environmental colonization and survival (see model in *Figure 8*). Notably, we demonstrated that both the capsule and the T6SS independently offer protection during antagonistic interactions with competitors, and these protective effects might be synergistic, at least under laboratory conditions (*Figure 8*). However, overproduction of the capsule in *A. baumannii* A118 impedes T6SS activity by hindering its assembly, potentially due to increased membrane tension from excess polysaccharides affecting membrane complex anchoring (*Figure 8*). This inhibition is alleviated in the double mutant Δ*bfmS*Δ*itrA*, which does not produce the capsule, indicating that the polysaccharide directly inhibits T6SS. Furthermore, capsule overproduction disrupts the organization of polysaccharides on the cell surface, likely impairing proper T6SS secretion (*Figure 8*). While the inhibition observed in Δ*bfmS* could result from both increased membrane tension and altered surface organization, our data conclusively show that strains lacking capsules exhibit enhanced T6SS activity and secretion compared20181905 to their encapsulated counterparts, suggesting that the capsule imposes steric hindrance on T6SS (*Figure 8*).

Interestingly, the inhibition of T6SS activity that we observed in the capsule-overexpressing mutant (Δ*bfmS*) also occurs under conditions of antibiotic-induced capsule overexpression. It is plausible that this inhibition is an adaptive response to withstand antibiotic exposure. Ultimately, this trade-off between T6SS functionality and capsule-mediated protection poses a competitive disadvantage, with the optimal balance achieved in the wild type, where both systems are functional.

Furthermore, we found that in secretion-impaired strains, the accumulation of Hcp is mitigated by a degradation mechanism involving the general ClpXP protease machinery (*Figure 8*). Given that the production of the T6SS involves the continuous synthesis and secretion of hundreds of protein components, we speculate that this degradation may serve as a strategy to alleviate proteotoxic stress and conserve energy, especially under unfavorable conditions such as antibiotic exposure.

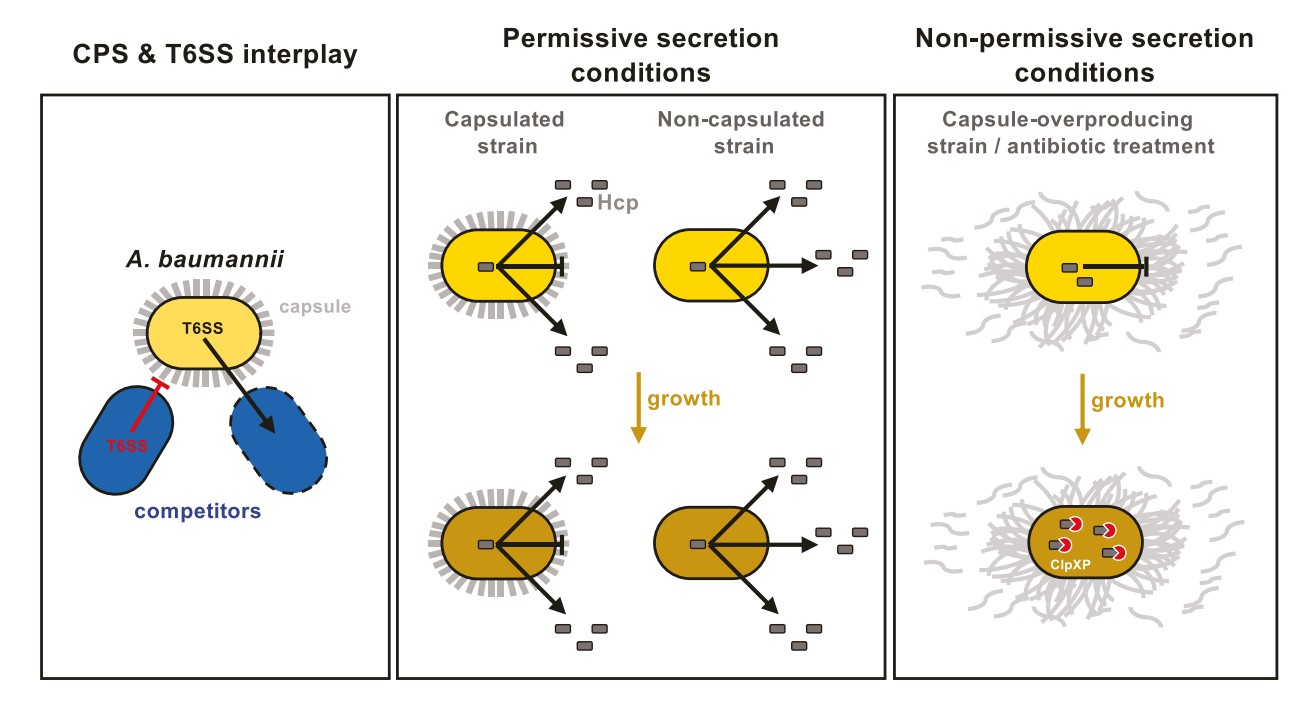

**Figure 8.** Model of capsule function and its impact on T6SS activity in *A. baumannii*. **Left panel:** *A. baumannii* capsular polysaccharide and T6SS protect against competitor T6SS attacks independently of immunity proteins. **Middle panel:** Under permissive conditions, the capsule only slightly impairs type VI secretion, preventing intracellular accumulation of Hcp. **Right panel:** Under non-permissive conditions, capsule overexpression—induced by *bfmS* deletion (Δ*bfmS*) or sub-MIC antibiotic treatment—modifies cell-associated polysaccharide organization, inhibiting T6SS secretion and therefore killing activity. ClpXP machinery degrades Hcp to prevent its accumulation over time.

In summary, this work establishes a foundational understanding of the interplay between extracellular polysaccharides, such as the capsule, and T6SS secretion process in *A. baumannii*. This interaction urges further characterization to develop effective strategies against this problematic and often antibiotic-resistant pathogen.

# Materials and methods

## Key resources table

| Reagent type (species) or resource | Designation | Source or reference | Identifiers | Additional information |
|---|---|---|---|---|
| Strain, strain background (*Acinetobacter baumannii*) | *Acinetobacter baumannii*, strain A118 | PMID:20181905 PMID:25164683 | *Acinetobacter baumannii* strain background | See *Supplementary file 1* for strains used in this study |
| Strain, strain background (*Escherichia coli*) | S17-1 λ pir | *Simon et al., 1983*, Nat Biotechnol **1**, 784–791 | *E. coli* cloning and mating strain | See *Supplementary file 1* for strains used in this study |
| Strain, strain background (*Escherichia coli*) | MFD*pir* | PMID:20935093 | *E. coli* mating strain | See *Supplementary file 1* for strains and plasmids used in this study |
| Commercial kit or assay | High-fidelity Q5 polymerase | New England Biolabs | M0491L | Polymerase for PCR |
| Commercial kit or assay | GoTaq G2 DNA polymerase | Promega | M7848 | Polymerase for PCR |
| Recombinant DNA reagent | Plasmids | This study | Plasmids | See *Supplementary file 1* for plasmids used in this study |

*Continued on next page*

*Continued*

| Reagent type (species) or resource | Designation | Source or reference | Identifiers | Additional information |
|---|---|---|---|---|
| Sequence-based reagent | PCR primers | This study | PCR primers | See *Supplementary file 2* for oligonucleotides used in this study |
| Antibody | Anti-Sigma70 monoclonal antibodies (Mouse IgG3) | BioLegend | 663208 | 1:10,000 diluted |
| Antibody | Anti-BSA HRP-conjugated antibodies (mouse monoclonal IgG$_1$) | Santa Cruz Biotechnology Inc | sc-32816 HRP | 1:2000 diluted |
| Antibody | Anti-GFP mouse monoclonal antibodies | Roche | 1181446001 | 1:5000 diluted |
| Antibody | Anti-rabbit IgG monoclonal antibodies conjugated to horseradish peroxidase (produced in goat) | Sigma-Aldrich | A9169 | 1:10,000 diluted |
| Antibody | HRP-conjugated anti-mouse antibodies (produced in sheep) | Sigma-Aldrich | A6782 | 1:20,000 diluted |
| Antibody | Anti Hcp-A118-2 polyclonal antibodies (produced in rabbit) | Eurogentec | Custom-made (*2110850 cl.2*) | 1:667 diluted |
| Other | CHROMagar *Acinetobacter* | CHROMagar, France | AC092 | Selective medium; additional, standard growth media are described under growth conditions |

## Bacterial strains, plasmids, and growth conditions

The bacterial strains and plasmids utilized in this study are detailed in *Supplementary file 1*. Generally, bacteria were grown in lysogeny broth (LB-Miller; Carl Roth, Switzerland) or on LB agar plates, aerobically at 37°C. *E. coli* strains S17-1 $\lambda$ *pir* (*Simon et al., 1983*) and MFD*pir* (*Ferrières et al., 2010*) served for cloning or as donors in mating experiments. For induction of the $P_{BAD}$ promoter, L-arabinose was added to the medium at a final concentration of 2%. Antibiotics and supplements were added as needed: kanamycin (50 µg/ml), carbenicillin (100 µg/ml), streptomycin (100 µg/ml), apramycin (100 µg/ml), chloramphenicol (5 µg/ml), gentamicin (15 µg/ml), and diaminopimelic acid (0.3 mM DAP). Isopropyl β-D-1-thiogalactopyranoside at a final concentration of 5 mM was added to the culture to induce gene expression from derivatives of the plasmid pMMB67EH.

## Genetic engineering of *A. baumannii*

DNA manipulations adhered to established molecular biology protocols, using enzymes as per the manufacturer's directions. Enzymes were purchased from these companies: High-fidelity Q5 polymerase (New England Biolabs), GoTaq polymerase (Promega), T4 DNA ligase (New England Biolabs), and restriction enzymes (New England Biolabs). Engineered strains and plasmids underwent initial PCR screening and were finally validated by Sanger sequencing of PCR-amplified fragments or plasmids.

*A. baumannii* mutants were created via allelic exchange with the counter-selectable suicide plasmid pGP704-Sac-Kan (*Metzger et al., 2019*; *Vesel and Blokesch, 2021*). Briefly, deletion constructs or *msfGFP* gene fusions were crafted to include >800 bp up- and downstream the target gene. These segments were amplified via PCR using oligonucleotides with 5′ restriction sites for later digestion. After enzymatic digestion, the fragments were ligated with the similarly cut pGP704-Sac-Kan plasmid using T4 DNA ligase and then introduced into chemically competent *E. coli* S17-1 $\lambda$ *pir* cells for further processes. Transformants were confirmed via colony PCR, and plasmid accuracy was ensured through Sanger sequencing. These plasmids were then introduced into *A. baumannii* through biparental mating for 8 hr at 37°C. Selection of single-crossover transconjugants utilized CHROMagar *Acinetobacter* (CHROMagar, France) plates or LB agar with chloramphenicol and kanamycin. After mating, the transconjugants were incubated at 37°C for 16 hr and then underwent selection at room temperature for the SacB-containing suicide plasmid's loss using plates of NaCl-free LB agar containing 10%

sucrose. Colony checks for antibiotic sensitivity confirmed plasmid loss. Mutants were validated through colony PCR and Sanger sequencing.

Selective mutants with antibiotic resistance markers were created via natural transformation, a method detailed in prior studies (*Godeux et al., 2020*; *Vesel et al., 2023*). The transforming material, generated by overlapping PCR, included a kanamycin resistance cassette (*aph*) flanked by FRT sites and 800 bp of homologous regions, enabling efficient transformation. Selected transformants on LB agar with kanamycin underwent verification through colony PCR and Sanger sequencing. The resistance cassette was then excised using the FLP/FRT recombinase system (*Tucker et al., 2014*), with the process and loss of the recombinase plasmid confirmed by antibiotic resistance tests, colony PCR, and Sanger sequencing, ensuring precise genetic manipulations.

## Interbacterial killing

The interbacterial killing assay was slightly modified from prior work (*Flaugnatti et al., 2021*). Bacteria were incubated overnight at 37°C with continuous shaking. They were then diluted 1:100 in fresh LB medium and grown until the optical density at 600 nm ($OD_{600}$) reached 1. For stationary-phase samples, overnight cultures after 15–16 hr of growth were used directly. Bacterial cultures (1 ml) were concentrated to an $OD_{600}$ of 10 with sterile PBS buffer. Attackers and prey were mixed in 1:1 or 1:5 ratios and spotted onto filters placed on LB agar plates. After incubation at 37°C for 4 hr, bacteria were resuspended in PBS, serially diluted, and spotted on selective media for an overnight incubation at 37°C. *A. baumannii* was selected on CHROMagar *Acinetobacter* medium (CHROMagar, France), while *E. coli* cells were selected on LB agar supplemented with streptomycin. Recovered colonies were counted to calculate the number of colony-forming units (CFU) per ml.

In this interbacterial killing assay to stimulate capsule production via chloramphenicol, bacteria were initially grown in LB for 20 hr at 37°C, then 1:100 diluted and grown further for 20 hr in LB without or with chloramphenicol (20 μg/ml). The cultures were subsequently processed as outlined above and the mixture of attackers and treated prey was spotted onto LB agar, with or without chloramphenicol (25 μg/ml), and incubated at 37°C for 4 hr. After incubation, bacteria were resuspended in PBS, diluted, and plated on selective media for overnight growth at 37°C, as mentioned above. Each experiment was repeated three independent times. Statistical significance was determined based on log-transformed data, with detection limits defined by the absence of at least one recoverable prey bacterium.

## Hcp secretion assay

To assess Hcp secretion, bacteria were grown in LB medium overnight at 37°C, followed by a 1:100 dilution and further aerobic cultivation until reaching an $OD_{600}$ of 1. For stationary-phase studies, overnight growth was extended to 15–16 hr before proceeding with further analyses.

Chloramphenicol-treated samples underwent a similar initial growth phase for 20 hr, followed by additional growth in the presence of chloramphenicol (20 μg/ml) for 20 hr, maintaining the same aerobic growth conditions. Subsequently, 2 ml of the culture was collected through centrifugation (5 min, 8000 rpm) and the supernatant filtered (0.22 μm filter; VWR). Secreted proteins in the supernatant were then precipitated using 10% ice cold trichloroacetic acid on ice for 2 hr. To verify consistent precipitation, BSA (100 μg/ml) was added to the supernatant before precipitation. The precipitated proteins were washed with 100% acetone, resuspended in 2× Laemmli buffer (50 μl/OD unit of initial culture), and heated before undergoing SDS–PAGE and western blot analysis.

## SDS–PAGE and western blotting

Proteins were separated on 12% mini-protean TGX stain-free precast gels and transferred to a PVDF membrane using the Trans-blot system as per the manufacturer's instructions (Bio-Rad). Membranes were blocked in 2.5% skim milk at room temperature for 30 min. Primary antibodies were raised in rabbits against synthetic peptides of Hcp (Eurogentec) and used at a dilution of 1:667 in 2.5% skim milk. After 1.5 hr of incubation, the membranes were washed three times with TBST (Tris-Buffered Saline with 0.1% Tween-20) buffer. They were then incubated for 1 hr with an anti-rabbit IgG conjugated to horseradish peroxidase (HRP) (A9169; Sigma-Aldrich) as the secondary antibody at a dilution of 1:10,000. Following three additional washes, the membranes were treated with Lumi-Light[PLUS] Western Blotting substrate (Roche, Switzerland) for signal development and visualized using

a ChemiDoc XRS +station (Bio-Rad). The anti-Sigma70 antibodies (BioLegend, USA distributed via Brunschwig, Switzerland) were used at a dilution of 1:10,000 to serve as a loading control in the experiment. Precipitated BSA was identified with anti-BSA-HRP-conjugated antibodies (Santa Cruz Biotechnology Inc), diluted at 1:2000, to verify the precipitation efficiency.

TssB-GFP fusions were identified using anti-GFP mouse monoclonal antibodies (1181446001; Roche) at a 1:5000 dilution, with HRP-conjugated anti-mouse antibodies (A6782; Sigma-Aldrich) at 1:20,000 as secondary antibodies for 1 hr.

## Serum killing assay

Bacteria were cultured overnight in 3 ml of LB medium under aerobic conditions at 37°C, then diluted 1:100 in fresh LB (2 ml) until the $OD_{600}$ reached 1. Following centrifugation, 1 ml of culture was washed and resuspended in PBS adjusted to an $OD_{600}$ of 1. For the assay, 40 µl of this bacterial suspension was mixed with 60 µl of baby rabbit complement serum (AbD Serotec) and incubated for 1 hr at 37°C. Controls included PBS (mock) and heat-inactivated (HI) serum, which was heated at 56°C for 30 min. The reactions were stopped by cooling the samples on ice, and surviving bacteria were quantified by plating serial dilutions on LB agar plates, incubated overnight at 37°C.

## Epifluorescence microscopy

After growing bacteria aerobically in 2 ml LB medium at 37°C to an $OD_{600}$ of 1, they were applied to agarose pads (1% agarose dissolved in 1× PBS) mounted on glass slides and covered by a coverslip. Cell visualization was performed using a Zeiss LSM 700 inverted confocal microscope (Zeiss, Switzerland) equipped with a fluorescence light source (Illuminator HXP 120), an AxioCam MRm high resolution camera, and controlled by the Zeiss Zen software (ZEN blue edition). Image analysis was conducted with Fiji software (2.0.0-rc-69/1.53f/Java 1.8.0_202 (64-bit); *Schindelin et al., 2012*). The displayed images are representative of three independent biological replicates.

## Quantification of T6SS sheath structures

In the pre-processing stage, acquired images were corrected for both drift and photobleaching. Drift was adjusted using the linear stack alignment with the SIFT plugin in ImageJ, based on phase contrast images (*Lowe, 2004*). To compensate for photobleaching, a histogram matching method was applied to the fluorescence channel (*Miura, 2020*).

For sheath structure detection, the pre-processed fluorescence images were analyzed using ilastik software (*Berg et al., 2019*) to create two types of classifiers: a pixel classifier for identifying T6SS-positive pixels and an object classifier for categorizing T6SS objects as either dotted shaped (contracted) or rod-shaped (extended). These classifiers were made by manually annotating a set of images representative of the dataset variability. However, due to challenges in differentiating between contracted and extended sheath structures, this distinction was not made in the final analysis.

For bacterial segmentation, since they remain stationary and unchanged in shape throughout the acquisition, the first phase contrast time-point was utilized. The segmentation method has been previously published (*Proutière et al., 2023*).

Data analysis involved pre-processing, segmentation, and sheath structure classification steps performed in ImageJ/Fiji using a Groovy script for batch processing (WorkFlow_File.groovy). This script generated a new multi-channel time-lapse stack per image, which consisted of the drift-compensated phase contrast channel, the bleach and drift-compensated fluorescent channel, a color-coded mask channel for contracted and extended sheath structures, and a label image of detected bacteria. The resulting stacks were used for visual assessment of the method and for downstream data analysis. A second script (CountObject_File) quantified the sheath structures per bacterium, per time point, and for each condition, output the data in a tsv file. All scripts, models, and classifiers were deposited on Zenodo (see data availability section).

## Extraction of CPS

Polysaccharide samples from cell lysates (membrane-bound) and culture supernatants (membrane-unbound) were prepared using a procedure slightly modified from previous studies (*Geisinger and Isberg, 2015*; *Tipton and Rather, 2019*). Briefly, bacteria grown overnight on LB agar plates at 37°C were resuspended in LB medium and adjusted to an $OD_{600}$ of 10. Cells were separated from the

supernatant by centrifugation. The supernatant was then precipitated with 75% ethanol at –20°C overnight. Meanwhile, the cell fraction was resuspended in lysis buffer (60 mM Tris, pH 8, 10 mM MgCl$_2$, 50 µM CaCl$_2$ with 20 µl/ml DNase and 3 mg/ml lysozyme) and incubated at 37°C for 1 hr. Post-vortexing, the cell fraction underwent three freeze–thaw cycles between liquid nitrogen and 37°C. The suspension was treated with 0.5% SDS for 30 min at 37°C, boiled at 100°C for 10 min, then treated with proteinase K (2 mg/ml) at 60°C for 1 hr. Following centrifugation (2 min, 15,000 × $g$), the supernatant was precipitated with 75% ethanol at –20°C overnight. The precipitated polysaccharides were centrifuged (30 min, 15,000 × $g$), resuspended in 40 µl 2× Laemmli buffer (Sigma-Aldrich, Switzerland), heated at 95°C for 10 min, and analyzed by SDS–PAGE and Alcian blue staining.

Samples treated with chloramphenicol were grown for 20 h in LB medium, followed by additional growth in LB medium supplemented with varying concentrations of chloramphenicol (0, 2.5, 5, 7.5, 10, or 20 µg/ml). Polysaccharides from the culture supernatants were precipitated and processed as explained above.

### Alcian blue staining

Precipitated polysaccharide samples were run on 12% mini-protean TGX stain-free precast gels (Bio-Rad) and stained with Alcian blue (0.1% Alcian blue in 40% ethanol, 60% 20 mM sodium acetate [pH 4.75]) for 1 hr, then destained overnight (40% ethanol, 60% 20 mM sodium acetate [pH 4.75]) (*Karlyshev and Wren, 2001*).

### Capsule visualization by TEM

For TEM visualization of CPS, previously published protocols were followed with minor modifications (*Chin et al., 2018*; *Valcek et al., 2023*). Briefly, the bacterial strains were cultured in LB medium for 15–16 hr at 37°C under shaking conditions, then a 500-µl sample was centrifuged to form a small pellet. This pellet was fixed on ice for 20 min with a mixture containing 2% paraformaldehyde and 2.5% glutaraldehyde in 0.1 M sodium cacodylate buffer (pH 7.4) with 1.55% L-lysine acetate and 0.075% ruthenium red. Following this, the fixed bacteria were washed three times in 0.1 M sodium cacodylate buffer (pH 7.4) with 0.075% ruthenium red. A second round of fixation was performed in the same fixation solution minus the lysine acetate for 2 hr. This fixation was followed by two additional rounds of washing in sodium cacodylate/ruthenium red buffer and then staining with 1% osmium tetroxide and 0.075% ruthenium red in 0.1 M cacodylate buffer for 1 hr at room temperature. Finally, the sample was washed with 0.075% ruthenium red in 0.1 M cacodylate buffer followed by distilled water and then dehydrated in a graded ethanol series before being embedded in an epon resin (Embed 812 embedding kit, EMS), which was polymerized for 24 hr at 60°C.

After hardening, 50 nm sections were prepared with a Leica UC7 Ultramicrotome and collected onto single-slot copper grids with a pioloform support film. The sections underwent contrasting for enhanced visibility with 2% lead citrate and 1% uranyl acetate and were imaged using a TEM (FEI Spirit) with a CCD camera (FEI Eagle) to capture each cell's structure.

### Structural model of the Hcp hexamer

The structural model of the Hcp hexamer is based on the crystal structure of the Hcp1 protein of *A. baumannii* strain AB0057 (RCSB PDB; *Berman et al., 2000*) code 4W64 (*Ruiz et al., 2015*) and was visualized using Mol* Viewer 2.9.3 (*Sehnal et al., 2021*).

### Materials availability

Biological materials, including bacterial strains and plasmids listed in *Supplementary file 1*, are available from the corresponding author upon provision of an appropriate Biosafety permit and the required paperwork.

## Acknowledgements

The authors would be like to acknowledge current and former members of the Blokesch group for insightful discussions, with special thanks to Nina Vesel for advice on *A. baumannii* genetics and David W Adams for general scientific advice, and Charles Van der Henst, Graham Knott, and Christel Genoud for helpful suggestions on Transmission Electron Microscopy. The authors also acknowledge the technical assistance of Sandrine Stutzmann, Laurie Righi, and Candice Stoudmann, and recognize

Lisa Metzger for plasmid engineering. The authors thank Nicolas Chiaruttini from the BioImaging and Optics platform of EPFL for development of the T6SS quantification script. The authors are grateful to Marek Basler, Xavier Charpentier, and Bryan W Davies for sharing of *A. baumannii* strains or plasmids. The authors also acknowledge the use of ChatGPT for language editing. This work was supported by the Swiss National Science Foundation (grant numbers 407240_167061 and 310030_204335) and a Howard Hughes Medical Institute (HHMI) International Research Scholarship (grant number 55008726) attributed to MB.

## Additional information

### Funding

| Funder | Grant reference number | Author |
| --- | --- | --- |
| Schweizerischer Nationalfonds zur Förderung der Wissenschaftlichen Forschung | 407240_167061 | Melanie Blokesch |
| Schweizerischer Nationalfonds zur Förderung der Wissenschaftlichen Forschung | 310030_204335 | Melanie Blokesch |
| Howard Hughes Medical Institute | 55008726 | Melanie Blokesch |

The funders had no role in study design, data collection and interpretation, or the decision to submit the work for publication.

### Author contributions

Nicolas Flaugnatti, Conceptualization, Formal analysis, Investigation, Methodology, Writing - original draft; Loriane Bader, Investigation; Mary Croisier-Coeytaux, Investigation, Methodology; Melanie Blokesch, Conceptualization, Resources, Data curation, Formal analysis, Supervision, Funding acquisition, Validation, Writing - original draft, Project administration, Writing - review and editing

### Author ORCIDs

Nicolas Flaugnatti  https://orcid.org/0000-0002-6073-3340
Melanie Blokesch  https://orcid.org/0000-0002-7024-1489

### Decision letter and Author response

Decision letter https://doi.org/10.7554/eLife.101032.sa1
Author response https://doi.org/10.7554/eLife.101032.sa2

## Additional files

### Supplementary files

Supplementary file 1. Table containing information about the strains and plasmids used in the study.

Supplementary file 2. Table containing information on the oligonucleotides used in the study.

Supplementary file 3. Source data for all non-gel and non-WB data are provided in this file.

MDAR checklist

### Data availability

Imaging dataset: All scripts, models, and classifiers used for image analyses have been deposited on Zenodo (https://doi.org/10.5281/zenodo.11039744). All raw images used in this study have been deposited on Zenodo (https://doi.org/10.5281/zenodo.14386836). All other data are included in the manuscript, with source data provided in *Supplementary file 3*.

The following datasets were generated:

| Author(s) | Year | Dataset title | Dataset URL | Database and Identifier |
|---|---|---|---|---|
| Flaugnatti N, Bader L, Croisier-Coeytaux M, Melanie B | 2024 | Capsular Polysaccharide Restrains Type VI Secretion in Acinetobacter baumannii | https://doi.org/10.5281/zenodo.11039744 | Zenodo, 10.5281/zenodo.11039744 |
| Nicolas F | 2024 | Raw data for the study by Flaugnatti et al., 2024 | https://doi.org/10.5281/zenodo.14386836 | Zenodo, 10.5281/zenodo.14386836 |

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
