## [Editor Report]

This important study reveals that capsular polysaccharide (CPS) in Acinetobacter baumannii not only protects against type VI secretion system (T6SS) attacks but also impairs the bacterium's own T6SS efficacy. The evidence supporting these findings is compelling. This work will be of interest to researchers focusing on bacterial defense mechanisms and interbacterial competition.

---

## [Decision Letter]

[Editors' note: this paper was reviewed by Review Commons.]

---

## [Author Response]

**Rebuttal letter**

First and foremost, we thank the three reviewers for their positive appraisals of our work and their helpful comments. We are confident that we have addressed all points of critique, as outlined below. The reviewers' comments are in blue, and our responses are in black.

Reviewer #1 (Evidence, reproducibility and clarity (Required)):Summary:The transport of effector proteins across membranes from the producing bacterium into a target cell is at the core of bacterial secretion systems. How an additional layer in form of a capsule affects effector export and the susceptibility towards effector import is not fully understood. Here, Flaugnatti and colleagues combined bacterial genetics with phenotypic assays and electron microscopy to demonstrate a dual role of a bacterial capsule in preventing T6SS-mediated effector export and promoting protection from effector import by another bacterium's T6SS. The wide variety of methods used, complementation of the mutants, and validation of the findings across strains strengthen the author's conclusions. Although the main conclusions seem straight forward, the authors unravel the unexpected complexity underlying these phenotypes with strong mechanistic work. In brief, a capsule-deficient mutant (∆itra) is shown to assemble its T6SS similar to the WT, yet secretes more Hcp than the WT and is better in T6SS-mediated killing of other bacteria. A capsule-overproducing mutant (∆bfmS) shows both, a partial deficiency in T6SS assembly and an additional reduction in exported Hcp, and is worse in T6SS-mediated killing than the WT. A mutant with a capsule similar to WT and deficient in cell sensing (∆tslA) forms the least T6SS apparatuses and is yet better in T6SS-mediated killing than the overcapsulated mutant. Together, these data show an effect of the capsule on (i) T6SS apparatus assembly, (ii) effector export, (iii) effector import, and (iv) the need for clearance of accumulating non-secreted Hcp by ClpXP.The work on a clinical isolate of Acinetobacter tumefaciens and the data on an impaired T6SS activity on other cells by antibiotic-induced capsulation is a strong demonstration of the work's clinical relevance in addition to the findings' conceptual novelty.In my view, the manuscript is outstanding with very high quality of experimental data, very well written text and very clear presentation of the data in figures. A few minor comments and suggestions below that I think would strengthen the manuscript.

We thank the reviewer for their enthusiasm.

Major comment:OPTIONAL: Figure 4c/l. 320: Having an indirect effect of an antibiotic on T6SS activity by antibiotic-induced capsule formation is very intriguing and contributes to the clinical relevance of the overall findings. When I saw the data in Figure 4c, the graph instantaneously reminded me of the panel in Figure 2a, where a similar phenotype is observed by changing the predator:prey ratio in the absence of any antibiotic. The authors themselves comment on the possibility of antibiotic-induced, reduced predator growth (and thereby a change in predator:prey ratio) as a one factor impacting the phenotype here. I am wondering if this data could be strengthened or better disentangled to test more precisely if it is the antibiotic induced capsule formation per se that affects T6SS-mediated killing by A. baumanii in the presence of antibiotics. Would it help to take the bfmS mutant along as a control for direct comparison to see if antibiotic-induced capsule formation of the WT to similar levels of the mutant results in the same killing phenotype? Would it help to test for T6SS-mediated killing in the presence and absence of antibiotics at multiple predator:prey ratios? Could the effect of the antibiotic on A. baumanii growth be measured and considered when choosing the ratio at which the bacteria are mixed?

The point raised by the reviewer is very important. As we have stated in the manuscript, the capsule-induced production using antibiotics impacts the growth of *A. baumannii* and could therefore change the predator-prey ratio, potentially affecting the observed phenotype. However, the antibiotic is expected to equally impact the non-encapsulated ΔitrA strain, yet this strain maintains very strong T6SS killing activity in the presence of chloramphenicol.

However, to still address this valid point more directly, we nonetheless repeated the experiments with a different attacker:prey ratio (5:1), and included the ∆*bfmS* mutant as control, as suggested by the expert. The new data are presented in Figure 4C and support our initial claim.

Minor comments:1) Figure 1D, l. 155, I might have missed this, do the authors happen to have the numbers of E. cloacae as well? This would strengthen the claim on A. baumannii survival because of E. cloacae is being killed.

The reviewer is correct; we did not include the survival of *E. cloacae* in the initial manuscript due to technical reasons (counter-selection of *E. cloacae*). During revision, we repeated the experiment using *E. cloacae* strains carrying a plasmid conferring kanamycin resistance (new Figure 1-supplement 1C and D), which provided direct evidence that *E. cloacae* is indeed killed by *A. baumannii* in a T6SS-dependent manner.

2) Figure 2, I suggest to write out the species name of the prey in the box with the ratio. With E. cloacae being referred to in the previous figure and starting with similar letters than *E. coli*, I wasn't sure at first sight what E. c. refers to.

Thank you for the comment; we have revised the figure as suggested.

3) U*se of the term "T6SS activity" throughout the manuscript (e.g. l. 182, l. 187). I leave this up to the authors. To me, it seems like an umbrella term for the initial observation and I see that such a term can be very handy for the writing. I just would like to mention that the use of the term was not always intuitive to me and sometimes even a bit misleading. For example, l. 182 refers to "increased T6SS activity". As a reader, I only know about 'T6SS activity on other cells' or 'a T6SS-mediated effect on other cells' at this point. T6SS apparatus assembly/firing activity is tested for specifically later and it turns out to differ between mutants. By the time the term is used in the discussion, it captures multiple nuanced phenotypes described by then. The more precise definition of the term in l. 200 helped to capture what exactly is meant by the authors.*

We revised the manuscript to rephrase these sentences, now using the term ‘T6SS-secretion activity’ for Hcp secretion assays and ‘T6SS-mediated killing activity’ for killing experiments.

4) l. 198-199 "Collectively, our findings indicate that CPS does not hinder the secretion process of the T6SS or the consequent elimination of competing cells". I might be missing something, I cannot entirely follow this sentence. Didn't the authors just show that the CPS does hinder T6SS-mediated elimination of competing cells in panel 2A and less secreted Hcp in the encapsulated WT compared to the non-encapsulated mutant in panel 2B?

We thank the reviewer for this comment. We realize that the sentence wasn’t well phrased, resulting in confusion. What we meant was that the T6SS is functional regarding its T6SS-mediated killing and secretion in the WT strain, while we also showed that the non-capsulated strain kills and secretes more T6SS material in the supernatant. Thus, there seems to be a balance between capsule production and T6SS activity in the WT. We revised the sentence to better reflect this meaning.

5) l. 224, typo, "in"

We corrected this typo. Thank you.

6) Two connected comments: l. 338, Just a thought, I am wondering about the title of the section. After reading it a second time, I think it is technically correct. When reading it first, I was a bit confused when getting to the data because apparatus assmebly is impaired in the capsule-overproducing strain and although "preserved", doesn't the data indicate that there is less T6SS assembly in the bfmS mutant and that this might be because of less cell sensing and isn't this a main point that there is a difference in apparatus assembly in the capsule overproducing strain compared to WT (other than no difference in apparatus assembly in the strain without capsule)? To me it seems not fully acknowledged as a finding in the interpretation of the data that less cells of the bfmS mutant have a T6SS apparatus. Isn't that interesting? A title along the lines of "Capsule-overproducing strain has preserved sensory function and assembles less T6SS apparatuses" would have been more intuitive for me. l. 352, In case I didn't miss a reference to this data earlier in the manuscript, I am wondering if it would be worth mentioning the finding on the reduced apparatus assembly of the bfmS mutant earlier, together with Figure 3 already. At least a sentence that mentions already that there is more coming later. When I got to this line in the manuscript and read the findings on the apparatus assembly, I first needed to go back to figure 3 and look at the data there again in light of this finding. It is mentioned here on the side but I think very important for the interpretation of the phenotypic data of the bfmS mutant shown earlier, isn't it? The tslA mutant is used beautifully here.

We thank the reviewer for the suggestion and the kind comment about the ‘beautiful usage of the *tslA* mutant’. We now revised the title of the corresponding paragraph to make it more intuitive (“Capsule-overproducing strains assemble fewer T6SS but retain sensory function”).

Regarding the comment about mentioning the T6SS apparatus assembly defect in the *bfmS* mutant earlier, we respectfully disagree. While we agree that this point is important and can partially explain the difference in killing activity, we believe that showing it together with the *tslA* mutant (Figure 5) makes more sense and is easier for the reader to understand.

7) Discussion: optional comment. On the one hand, I like the concise discussion. On the other hand, I see more potential here for bringing it all together (potentially at the expense of shortening some of the introduction). I think the subtleties of the findings are complex. For example, I could envision a graphical summary with a working model on all the effects of a capsule on the T6SS and its potential clinical relevance making the study accessible to even more readers.

In the revised manuscript, we included a summarizing model as new Figure 8.

SignificanceGeneral assessment: I consider the story very strong in terms of novelty, experimental approaches used, quality of the data, quality of the writing and figures of the manuscript. In my view, the aspects that could be improved are optional/minor and concern only one figure and some phrasing.Advance: I see major advance in the findings (i, mechanistic) on the mechanism of how the capsule interferes with T6SS, (ii, fundamental) on the discovery of ClpXP degrading Hcp, and (iii, clinical) on the meaning of antibiotic treatment for the T6SS of this clinically relevant and often multi-drug resistant bacterial species, which strongly complements existing work on the T6SS and antibiotics in A. baumanii (e.g. of the Feldman group). As the authors write themselves, the starting points of the study of a capsule protecting from a T6SS and the effect of a T6SS on other cells being negatively impacted by a capsule were known, although not studied in one species and not understood mechanistically.Audience: I see the result of interest to a broad audience in the fields of bacteria-bacteria interactions, Acinetobacter baumanii, type VI secretion, antimicrobial resistance, bacterial capsules.

We once again thank the reviewer and highly appreciate their positive and constructive feedback on our work. We hope the reviewer will be satisfied with the revised version of our manuscript.

Reviewer #2 (Evidence, reproducibility and clarity (Required)):In the manuscript by Flaugnatti et al., the authors provide clear evidence of the interplay between capsule outer coat production and the Type VI secretion system (T6SS) in Acinetobacter baumannii. The authors demonstrate that the presence of the capsule or the activity of the T6SS enhances survival against attacking bacteria. However, they also show that in their model bacterium, the (over)production of the capsule likely hinders T6SS dynamics, thereby reducing overall killing efficiency. Additionally, they reveal that the amount of the T6SS component Hcp is regulated in cells that can no longer assemble and/or secrete via the T6SS, presumably by the ClpXP protease. Overall, the experiments are well designed, and most conclusions are supported by the data and appropriate controls. I have however some suggestions that could further strengthen the manuscript prior to publication.

We are grateful for the reviewer’s enthusiasm and implemented their comments and suggestions in the revised version of the manuscript.

Major comments:Line 164. The authors use *E. coli* as prey to test the T6SS activity of A. baumannii. Why not directly use the E. cloacae strain (with or without T6SS) for this purpose? This would provide direct evidence that A. baumannii uses its T6SS to kill E. cloacae, thus confirming the authors conclusions in this section.

We thank the reviewer for this comment. We used *E. coli* to assess the functionality of the T6SS in different strains of *A. baumannii*, as it is commonly done in the T6SS field. However, as suggested by reviewer 1 (see minor comment #1) and in response to this query, we also provided survival data of *E. cloacae* in the revised manuscript using plasmid-carrying *E. cloacae* derivatives that allowed direct selection. The new data is shown in Figure 1-supplement 1C and D.

In Figure 2, the authors show that a non-capsulated strain kills more effectively and secretes more than a WT, but has a similar number of T6SS. They suggest in their conclusion that "the observed increase in T6SS activity in the non-capsulated strain suggests a compensatory mechanism for the absence of the protective capsule layer." This conclusion implies the presence of an "active" regulatory mechanism that would increase the number of successful T6SS firing events, which has not been demonstrated. Could it not simply be that the capsule blocks some shots that cannot penetrate and are therefore ineffective? This hypothesis is mentioned in lines 204-208. The authors should clarify the conclusion of this section. Given the challenge this may pose in A. baumannii, I suggest that the authors quantify the assembly/firing dynamics of the T6SS under WT and ΔitrA conditions. This would help distinguish between the two hypotheses explaining better firing in non-capsulated cells: i.e., if the number of assembled T6SS is the same in both strains (Figure 2C and 2D), do non-capsulated cells assemble/fire faster, indicating an adaptation in regulation, or do we observe the same dynamics, suggesting a simple physical barrier blocking the passage of certain T6SS firing events?

We realize that the sentence, and more specifically the word "compensatory," might have been misleading and thank the reviewer for bringing this to our attention. What we meant to convey is that there is a balance between capsule production and T6SS activity; if disturbed, the balance shifts in one direction or the other. Specifically, there is more protection through the production of a thicker capsule (e.g., in the ∆*bfmS* mutant or under sub-MIC conditions of antibiotics, regulated by the Bfm system, as mentioned in the text) or more T6SS activity when less capsule is present (e.g., in the Δ*itrA* mutant, which we propose is caused by the lack of the steric hindrance). We rephrased this sentence in the revised manuscript to better convey this message.

Regarding the quantification of T6SS dynamic assembly/firing events between the capsulated (WT) and non-capsulated (ΔitrA) strains, we do not think this is required for this study, as the amount of secreted Hcp reflects the overall activity of the system. Importantly, we also do not have the technical means to quantify assembly/firing rates under Biosafety 2 conditions, as this requires specialized microscopes with very fast acquisition options (see, for instance, Basler, Pilhofer et al., 2012, Nature). Indeed, very few labs in the T6SS field have been able to measure such rates.

Line 428-429. It is mentioned that the deletion of lon does not have a notable effect. However, I observe that the absence of Lon alone causes a more rapid degradation of Hcp in the cells compared to the WT strain (Figure 7B). How do the authors explain that the absence of this protease (whether under conditions of Hcp accumulation or not) increases the degradation of this protein in the cell? This explanation should be included in the manuscript.

That’s a fair point. We didn’t address this point further, as the deletion of *lon* didn’t resolve the issue of why Hcp is degraded. In fact, the opposite seems to be the case, as there is less Hcp in the ∆*lon* strain compared to the WT. While this observation is not directly relevant to the question of why Hcp is degraded late during growth in secretion-impaired strains, we mentioned it in the revised manuscript, as suggested.

Please also note that a strong growth defect of a Δ*lon*Δ*clpXP* double mutant impaired further investigation in this direction.

Minor comments:Throughout the manuscript, the authors use the term "predator" to refer to A. baumannii. Predation is a specific phenomenon that involves killing for nourishment. To my knowledge, the T6SS has never been shown to be a predation weapon but rather a weapon for interbacterial competition, which is a different concept. If this has not been demonstrated in A. baumannii, the authors should replace the term "predator" with "attacker" (or an equivalent term) to clarify the context.

We thank the reviewer for this comment. The term “predator,” as highlighted by the reviewer, typically implies killing for nourishment/cellular products. In the context of T6SS, it facilitates the killing of competitors, releasing DNA into the environment that can subsequently be acquired through natural competence for transformation, as observed in species like *Vibrio cholerae* (our work by Borgeaud et al., 2015, Science) or other *Acinetobacter* species such as *Acinetobacter baylyi* (Ringel et al., 2017, Cell Rep.; Cooper et al., 2017, *eLife*). The acquisition of DNA reflects the killing for cellular products of the prey. As most *A. baumannii* strains are also naturally competent, this justifies the usage of the predator and prey nomenclature.

Apart from this fact, it seems to be a matter of nomenclature, with many papers in the field using one term or the other. Yet, ultimately, this doesn’t change any of the scientific findings. Therefore, to address this point of critique, we changed the term ‘predator’ to ‘attacker’ throughout the revised manuscript.

Line 274. Since the authors stated that in the Wzc mutant, the capsule is "predominantly found in the supernatant and only loosely attached to the cell," this result is not unexpected. This finding is also consistent with the previous results from Figure 3A and B, which show sensitivity to complement-mediated killing and the weak amount of (ab)normal CPS produced in that strain, further confirmed by Figure 3E

We fully agree with the reviewer’s suggestion and removed the statement from the revised manuscript.

Line 299. The authors speculate that "… T6SS may deploy through gaps akin to arrow-slits in the capsule's mesh…". However, this is very unlikely since a WT strain kills (Figure 3C) and secretes (Figure 2B and 3D) less effectively than the itrA mutant, suggesting that the T6SS is not assembled in the "right places" devoid of CPS; otherwise, we would expect similar T6SS activity. Based on the results in Figure 2 (and my earlier comment), this implies that A. baumannii assembles its T6SS randomly, and in the presence of the capsule, its shots would need to be in the right place to penetrate the envelope and reach the target. Could the authors comment on this point and provide a model figure to better visualize the interplay between the capsule and T6SS under the three major conditions: WT, non-capsulated, and capsule overproduction?

We thank the reviewer and agree with their comment. We discussed the hypothesis of T6SS deployment through gaps, drawing a parallel to what was proposed for biofilm/EPS and T6SS in *V. cholerae* (Toska et al., 2018, PNAS). To clarify our findings further, we now included a model summarizing our results as new Figure 8, as also requested by reviewer 1 (see minor comment #7).

In Figure 5A, the microscopy panels should be adjusted to the same dynamic range as the WT (which represents a true signal), which does not appear to be the case for the tlsA mutant panel for instance. The image gives the impression of a large amount of free TssB-msfGFP in the cytoplasm. However, this effect is due to the dynamic range being adjusted to display a signal. This observation is consistent with the fact that the amount of TssB-msfGFP protein is identical across all strains (Figure S2F).

We thank the reviewer for this comment. In fact, all images were adjusted to the same dynamic range as the WT in our initial submission. The observed differences in fluorescence intensity in Figure 5, which we confirmed by quantifying total fluorescence per cell (new Figure 5-supplement 1A), were also unexpected to us. Thus, during the revision, we consulted an expert from our microscopy facility, who confirmed that acquisition times and channel adjustments were identical for all samples. Given that TssB-msfGFP protein levels were consistent between *bfmS*-positive and *bfmS*-negative strains, we speculate that GFP folding might be affected in the ΔbfmS background.

Importantly, regardless of image presentation, the enumeration of T6SS structures remains consistent, with the “dimmer” Δ*bfmS*Δ*itrA* strain displaying WT-like T6SS sheath assembly, which was the primary focus of this experiment.

Unless I am mistaken, the authors do not comment on the fact that in a ΔbfmS strain, the number of T6SS is halved compared to a WT or ΔitrA strain. If capsule overproduction only partially limits the TslA-dependant T6SS assembly, how can this result be explained? Is it related to the degradation of Hcp in this background, which ultimately limits the formation of T6SS? If so, it would be interesting to mention this connection in the section "Prolonged secretion inhibition triggers Hcp degradation”We do not believe that the reduced T6SS assembly observed in the Δ*bfmS* background (Figure 5) is linked to Hcp degradation, as the latter process occurs only in the later growth phase (see Figure 6C). Therefore, we refrained from speculating this connection in the section ‘*Prolonged secretion inhibition triggers Hcp degradation’*.SignificanceThis work is highly intriguing as it not only delves into the specific mechanisms involved but also connects fundamental elements in bacterial competition, i.e., the necessity for self-protection and aggression for survival. The manuscript offers valuable insights into cellular dynamics at a microscale level and prompts new inquiries into the regulation of these systems on a population scale. The work is well-done and the writing is also clear. I am convinced that this work represents another significant step towards understanding bacterial mechanisms and will undoubtedly spark considerable interest in the field.

We sincerely thank reviewer #2 for their constructive comments, which significantly improved our manuscript.

Reviewer #3 (Evidence, reproducibility and clarity (Required)):The manuscript by Flaugnatti et al. investigates the relationship between functions of the T6SS in A. baumannii and production of capsular polysaccharide. The manuscript argues that (1) capsule protects A. baumannii against T6SS-mediated attack by other bacteria, (2) capsule also interferes with the bacterium's own T6SS activity, and (3) the T6SS inner tube protein Hcp is regulated by degradation by ClpXP. The main critiques regard the first two conclusions, which seem to be based solely on use of a mutant that has a confounding effect as described below; and to strengthen the third claim by further exploring the results of overexpressing Hcp and by determining whether there is a fitness benefit for Hcp regulation.

We thank reviewer #3 for their relevant input. We conducted additional experiments based on their comments, and these were incorporated into the revised manuscript.

Main items:Throughout the paper, an itrA deletion mutant is used as the capsule-deficient strain and conclusions are drawn about role of capsule based on this mutant. However, itrA deletion also eliminates the protein O-glycosylation pathway (Lees-miller et al. 2013), a potential confounder. Analysis of mutants specifically deficient in the high-molecular weight capsule but not protein glycosylation, and/or mutants in the protein o-glycosylation enzyme, should be incorporated into the study to enhance the ability to make conclusions about the role of the capsule.

Fair point. We thank the reviewer for this important suggestion. To distinguish between the *O*-glycosylation pathway and capsule production, we generate a ∆*pglL* mutant strain (specific to *O*-glycosylation), as suggested, and repeated the key experiments of the initial Figure 2. The new data are presented in the revised Figure 2 and conclusively indicate that *O*-glycosylation is not implicated in the observed phenotypes.

Evidence could be provided to support the idea raised in lines 482-483 that T6SS component accumulation is toxic ("degradation [of T6SS components] could serve as a strategy to alleviate proteotoxic stress…"). For example, growth curves of ∆clpXP strains with and without hcp could be analyzed, to determine how degrading Hcp is helping the bacteria.

As suggested by the reviewer, we performed growth curves of Δ*clpXP* strains with and without *hcp* (see Author response image 1) but did not observe any growth defects. However, standard growth curves under laboratory conditions may not capture natural conditions where minor fitness disadvantages could be detrimental. While investigating why Hcp is degraded under these conditions is beyond the scope of this study, we clarified in the revised manuscript that ‘alleviating proteotoxic stress’ remains speculative.

**Author response image 1. sa2fig1:** 

The possible ClpXP recognition sequence identified at the C terminus of Hcp is interesting-does overexpression of an Hcp variant lacking/altered in this motif alter its protein levels compared to WT Hcp?We thank the reviewer for this suggestion. We performed the recommended experiments, which yielded highly exciting results, and have included the data in the revised manuscript (Figure 7 and Figure 7-supplement 1). Briefly, we demonstrated that Hcp variants lacking the C-terminal 11 amino acids, or with the last two alanine residues replaced by aspartate, were no longer degraded in secretion-deficient *A. baumannii*.

Minor items:A better explanation could be provided for why overexpressing hcp in WT but not in ∆hcp leads to increased Hcp protein levels. There is a statement about Hcp being regulated post transcriptionally, possibly by degradation (lines 422-423), but would that not also result in regulation in the WT strain?

The reviewer is absolutely correct here. Despite careful genetic engineering, we believe that the *hcp* mutant used may have a polar effect, causing Hcp accumulation only in the ∆*hcp* + p-*hcp* strain but not in the WT + p-*hcp* strain, which remains capable of secretion. The ∆*hcp* strain therefore mimics the secretion-impaired *tssB* mutant. We clarified this fact in the revised manuscript.

An untreated control is needed in Figure 4B.

The untreated samples were shown in all previous figures. However, we understand the reviewer's point and repeated the experiment with the untreated control included in the same experiment (see revised Figure 4B).

line 179: please clarify "reflecting better invading bacteria"

We appreciate the reviewer mentioning this oversight. We meant to compare this to a situation where a bacterium invades an already existing community, resulting in an attacker-prey ratio below 1. We revised the manuscript accordingly.

line 351: consider rewording the statement that ∆tslA results in decreased in T6SS assembly and activity using the tssB-msfGFP microscopy assay; it is not clear that activity is measured in this assay.

The reviewer is correct. We revised the sentence accordingly to better reflect the T6SS assembly phenotype.

lines 260-265: This experiment could use clarifying, but it would seem that it requires analysis of the secreted capsule levels in the tssB mutant to show it does not produce extracellular capsule to the same extent that ∆bfmS does.

We thank the reviewer for the suggestion and included these experimental data in the revised manuscript (Figure 3-supplement 1D in the revised manuscript).

Figure 6C and 7A labelling could be improved to avoid potential confusion that the bar graphs are quantifying the western blot. E.g., could add a corresponding vertical label to the Western data, or consider changing "relative expression of hcp" to something reflecting analysis of transcript levels.

We improved these figures, as requested, by splitting the qPCR and Western blot data into independent panels for the main figures.

lines 416-417 and Figure 7A: states that "hcp mRNA levels increased significantly", but more careful wording could be used because the WT's transcript change is not significant after overexpression (though it is significant in ∆hcp).

Point well taken. We revised the sentence to make its meaning unambiguous.

lines 479-480 states that in secretion-impaired strains accumulation of Hcp is mitigated by ClpXP; while this was shown for ∆tssB, was this also the case for ∆bfmS?

This is an interesting suggestion. Accordingly, we generated the Δ*bfmS*Δ*clpXP* double mutant and tested its Hcp levels along with the necessary controls, which mirrored the Δ*tssB* situation. The data are included in the revised manuscript (Figure 7—figure supplement 1B).

SignificanceThe strengths of the study are the focus on a clinically significant pathogen, the potential novel roles for the important capsule virulence factor of A. baumannii, and the identification of novel points of control of the T6SS. The analyses of T6SS function are thorough and carefully performed.

We thank the reviewer for their comments, which significantly strengthened our work, particularly regarding the capsule aspect.